# High residual carriage of vaccine-serotype *Streptococcus pneumoniae* after introduction of pneumococcal conjugate vaccine in Malawi

Todd D. Swarthout [1,2,3✉], Claudio Fronterre [4], José Lourenço [5], Uri Obolski [6,7], Andrea Gori[2], Naor Bar-Zeev [1,8], Dean Everett [1,9], Arox W. Kamng'ona [10], Thandie S. Mwalukomo[11], Andrew A. Mataya[1], Charles Mwansambo[12], Marjory Banda[13], Sunetra Gupta[5], Peter Diggle[4], Neil French[1,14,15] & Robert S. Heyderman[1,2,15]

There are concerns that pneumococcal conjugate vaccines (PCVs) in sub-Saharan Africa sub-optimally interrupt *Streptococcus pneumoniae* vaccine-serotype (VT) carriage and transmission. Here we assess PCV carriage using rolling, prospective nasopharyngeal carriage surveys between 2015 and 2018, 3.6–7.1 years after Malawi's 2011 PCV13 introduction. Carriage decay rate is analysed using non-linear regression. Despite evidence of reduction in VT carriage over the study period, there is high persistent residual carriage. This includes among PCV-vaccinated children 3–5-year-old (16.1% relative reduction from 19.9% to 16.7%); PCV-unvaccinated children 6–8-year-old (40.5% reduction from 26.4% to 15.7%); HIV-infected adults 18-40-years-old on antiretroviral therapy (41.4% reduction from 15.2% to 8.9%). VT carriage prevalence half-life is similar among PCV-vaccinated and PCV-unvaccinated children (3.26 and 3.34 years, respectively). Compared with high-income settings, there is high residual VT carriage 3.6–7.1 years after PCV introduction. Rigorous evaluation of strategies to augment vaccine-induced control of carriage, including alternative schedules and catch-up campaigns, is required.

[1] Malawi-Liverpool-Wellcome Trust Clinical Research Programme, Blantyre, Malawi. [2] NIHR Global Health Research Unit on Mucosal Pathogens, Division of Infection and Immunity, University College London, London, UK. [3] Clinical Sciences Department, Liverpool School of Tropical Medicine, Liverpool, UK. [4] CHICAS, Lancaster Medical School, Lancaster University, Lancaster, UK. [5] Department of Zoology, University of Oxford, Oxford, UK. [6] School of Public Health, Tel Aviv University, Tel Aviv, Israel. [7] Porter School of the Environment and Earth Sciences, Tel Aviv University, Tel Aviv, Israel. [8] International Vaccine Access Center, Department of International Health, Johns Hopkins University, Baltimore, USA. [9] The Queens Medical Research Institute, University of Edinburgh, Edinburgh, UK. [10] Department of Biomedical Sciences, College of Medicine, University of Malawi, Blantyre, Malawi. [11] Department of Medicine, College of Medicine, University of Malawi, Blantyre, Malawi. [12] Ministry of Health, Lilongwe, Malawi. [13] Ministry of Education, Blantyre, Malawi. [14] Centre for Global Vaccine Research, Institute of Infection and Global Health, University of Liverpool, Liverpool, UK. [15]These authors contributed equally: Neil French, Robert S. Heyderman. ✉email: t.swarthout@ucl.ac.uk

*S*treptococcus pneumoniae is estimated to be responsible for over 318,000 (uncertainty ratio: 207,000–395,000) deaths every year in children aged 1–59 months worldwide, with the highest mortality burden among African children[1]. *S. pneumoniae* has over 90 immunological serotypes and is a common coloniser of the human nasopharynx, particularly in young children, resource-poor and HIV-affected populations[2]. Although most carriers are asymptomatic, pneumococcal colonisation is a necessary prerequisite for transmission and the development of disease, including pneumonia, meningitis, and septicaemia[3].

In high-income country (HIC) settings (including Europe and North America), routine infant administration of pneumococcal conjugate vaccine (PCV) has rapidly reduced vaccine serotype (VT) invasive pneumococcal disease (IPD)[4–6] and carriage[7–9]. Importantly, this has occurred in vaccinated and unvaccinated age groups. Thus, indirect protection resulting from a reduction in carriage and transmission amplifies PCV impact and cost-effectiveness[10]. In HICs, PCV was commonly initiated soon after becoming available in 2000, initially with 7-valent Prevenar® (PCV7) usually with a booster dose in the second year of life in addition to doses in infancy (e.g. 2 + 1 or 3 + 1), subsequently switching to 10-valent Synflorix® (PCV10) or 13-valent Prevenar13® (PCV13). PCVs in Africa were predominantly introduced from 2011 as supportive funding became available through the Global Alliance for Vaccines and Immunisation (GAVI, the Vaccine Alliance). The schedules most often implemented, with PCV10 or PCV13, were given in infancy without a booster (3 + 0). South Africa, Rwanda, and The Gambia were the exceptions, implementing PCV7 in 2009 and then PCV13. South Africa has used a 2 + 1 schedule, with the third dose given at 9 months of age with the measles vaccine (See Supplementary Table 1 for an overview of the regional differences in vaccine schedules and formulations).

Pneumococcal epidemiology in sub-Saharan Africa (sSA) is characterised by higher rates of carriage and transmission than HIC settings[11–15]. Carriage studies pre-dating PCV introduction in Kenya[11], Mozambique[12], Malawi[13], The Gambia[14], and South Africa[15], for example, reported pneumococcal carriage prevalence values ranging from 59 to 90% among children < 5 years old, with colonisation occurring rapidly early in life[16]. This differs markedly from high-income settings including, for example, the UK[17], the USA[18], and the Netherlands[19], reporting pre-PCV pneumococcal carriage prevalences of 48–68%.

Vaccine trials and post-routine-introduction studies in sSA have demonstrated substantial direct effects of PCV against IPD, pneumonia, and all-cause mortality among young children[20–23]. Though PCV impact is not directly comparable between countries, and regions given differences in PCV implementation strategies (formulations and schedules), these countries (e.g., Malawi [PCV13; 3 + 0][13], The Gambia [PCV7 and PCV13; 2 + 1][20,21], Kenya [PCV10; 3 + 0][22], South Africa [PCV7 and PCV13; 2 + 1][23,24], and Mozambique [PCV13; 3 + 0][25],) have consistently reported higher residual VT carriage than HICs (United Kingdom [PCV7 and PCV13; 2 + 1][17], Netherlands [PCV7 and PCV10; 3 + 1 and 2 + 1][26], USA [PCV7 and PCV13; 3 + 1][27]). See Supplementary Table 2 for an overview of regional differences in residual VT carriage prevalence after the introduction of PCV into routine Extended Programmes on Immunisation [EPIs]).

As reported for HIC settings[17,27,28], there is also increasing evidence of non-vaccine-serotype (NVT) carriage replacement in sSA[29,30]. While NVT typically cause less invasive disease[31], capsule switching[32] and genotypic shifts[33] may lead to increases in replacement. This is of particular concern in many sub-Saharan African countries where the 3 + 0 schedule has been implemented into infant EPIs[34].

In November 2011, Malawi (previously PCV naïve) introduced PCV13 as part of the national EPI, using a 3 + 0 schedule (6, 10, and 14 weeks of age). A three-dose catch-up vaccination campaign included infants < 1 year of age at the date of first dose, receiving three doses at 1-month intervals. Previous field studies in Malawi have reported high PCV13 uptake rates of 90–95%[35,36], similar to the 92% PCV-3 coverage recently reported by WHO/UNICEF[37]. A 2015–2016 Malawian Ministry of Health assessment reported a 12.8% HIV prevalence among adult women (aged 15–64) and an HIV prevalence of 8.2% among adult men[38]. In 2011, Malawi adopted Option B+, whereby all HIV-infected pregnant or breastfeeding women commence life-long full antiretroviral therapy (ART) regardless of clinical or immunological stage, dramatically reducing mother-to-child transmission[39].

An earlier, small community-surveillance study in Karonga District, northern Malawi, compared pneumococcal carriage before and 2 years after PCV introduction[13]. VT carriage among young PCV-vaccinated children (1–4 years of age) was 28.2% before vs 16.5% after PCV introduction (Supplementary Table 5). These data led us to hypothesise that despite evidence of PCV13 impact on IPD and pneumonia in Malawi[40,41], in the longer term after vaccine introduction there would be persistently high residual VT carriage and that this would maintain transmission in both childhood and adult reservoirs. In this large population-based study, we investigate this among children PCV13-vaccinated through the routine EPI (in whom vaccine-induced immunity begins to wane after the first year of life[42]); children too old to have received PCV13; and HIV-infected adults on ART who do not routinely receive pneumococcal vaccination but were previously demonstrated to have a high carriage prevalence[43,44]. We find that, despite evidence of reduction in VT carriage since PCV13 introduction, there is high persistent residual carriage among PCV-vaccinated and PCV-unvaccinated study populations. We show a VT carriage prevalence half-life that is similar among older PCV-vaccinated and PCV-unvaccinated children. These results, not dissimilar to a number of other sub-Saharan African countries, underline the need for rigorous evaluation of strategies to augment vaccine-induced control of carriage.

## Results

**Recruitment**. Between 19 June 2015 and 6 December 2018, seven rolling carriage surveys were completed: dates for each survey were, respectively, (1) June to August 2015; (2) October 2015 to April 2016; (3) May to October 2016; (4) November 2016 to April 2017; (5) May to October 2017; (6) November 2017 to June 2018; (7) June to December 2018. This spanned a period of 3.6–7.1 years after Malawi's 12 November 2011 introduction of PCV. Given that seasonality was found to be associated with pneumococcal carriage in a previous study in Malawi[16], enrolment was evenly distributed across seasons of high and low carriage incidence.

Overall, 7554 individuals were screened (Fig. 1), including 371 infants 4–8 weeks old (prior to receiving the first dose PCV), 602 PCV-vaccinated children 18 weeks to 1 year old, 538 PCV-vaccinated children 2 years old, 2696 PCV-vaccinated children 3–7 years old, 1505 PCV-unvaccinated children 3–10 years old, and 1842 HIV-infected adults 18–40 years old and on ART (PCV-unvaccinated). Among those screened, 24 (6.5%) infants, 196 (5.1%) children age-eligible for PCV, 96 (6.4%) children age-ineligible for PCV, and 67 (3.6%) HIV-infected adults were excluded (Fig. 1) from recruitment after screening. Twenty-three participants (eighteen children, five HIV-infected adults) did not allow a swab to be collected after recruitment. The final analysis included 7148 participants: 346 infants recruited before receiving

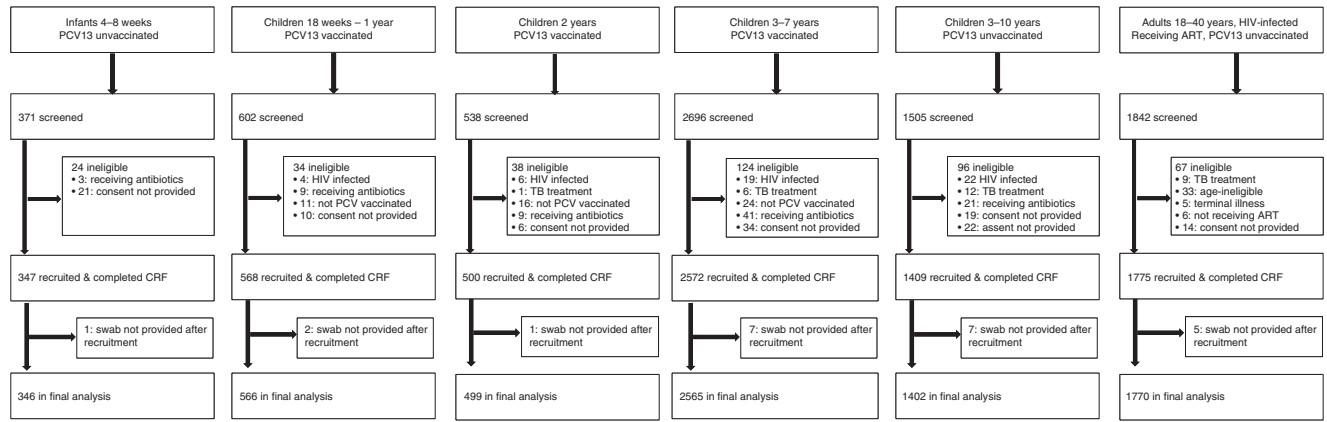

**Fig. 1 Recruitment flow diagram.** Among school-goers (6–10 years old), the number reported as screened does not include parents/guardians who did not come to the school after written invitation. Overall, 3781 letters of invitation were sent to parents/guardians during recruitment of school-goers. Of these, 1493 (39.5%) came to the school to be further informed of the study and consider their child's participation. Reasons for not accepting the invitation were not routinely collected. A total 1427 school-goers were recruited. Among children recruited from household (18 weeks to 5 years old), study teams maintained a diary of number of homes visited (i.e. knocked on gate/door of main house). An average of 7.2 household were approached for every child screened. Reasons for failing to screen-at-household included, (i) no one home, (ii) someone home but no age-eligible child home, and (iii) age-eligible child home but no parent/guardian available.

their first dose PCV, 566 children 18 weeks to 1 year old and PCV vaccinated, 499 children 2 years old and PCV vaccinated, 2565 children 3–7 years and PCV vaccinated, 1402 children 3–10 years old and PCV-unvaccinated, and 1770 HIV-infected adults on ART and PCV-unvaccinated. Among school-goers, of the 3781 invitation letters sent home, 1493 (39.5%) resulted in a parent/guardian visiting the school for further study information and screening. The number of letters actually received by the parents/guardians and the reasons for parents/guardians not accepting the invitations were not routinely documented. During household recruitment, study teams maintained a diary to record the number of homes visited. An average of 7.2 households was approached for every child screened. Reasons for failing to screen-at-household included, (i) no one home, (ii) someone home but no age-eligible child available, and (iii) age-eligible child home but no parent/guardian available. Among the children in the final analysis, 3605 were recruited from households and 1427 from schools.

**Demographics and vaccination history.** When assessing the distribution of demographics in Table 1 to optimise the overlap in ages, we restricted the analysis to children 3–7 years of age in each PCV-unvaccinated and PCV-vaccinated children. Though the surveyed groups had similar demographics, a higher proportion of older PCV-unvaccinated children (recruited from schools) lived in houses with some more costly infrastructure (walls of burnt brick, a water toilet, a water tap to house); these children also had a higher index of household possessions and a higher crowding index. Among those screened and age-eligible for PCV vaccination, 98.7% (3785/3836) reported receiving at least one dose of PCV. Among the 3630 PCV-vaccinated children recruited and providing a nasopharyngeal swab (NPS), 1209 (33.3%) had documented (in the health passport) vaccination status with vaccination dates; ranging from 86.5% among the youngest vaccinated age group (18 weeks to 1 year old) to 25.7% among the oldest vaccinated age group (3–7 years old). Among those with health passports confirming dates of vaccination, the median (IQR) ages at first, second, and third dose of PCV were 6.3 (3.2), 11.2 (5.0), and 16.4 (8.1) weeks, respectively; 1143 (94.5%) received three doses PCV, 24 (2.0%) only two doses, and 42 (3.5%) only one dose.

**Pneumococcal carriage.** Figure 2 reports the carriage dynamics of *S. pneumoniae* per rolling survey, stratified by study group (refer to Supplementary Tables 3 and 4 for VT and NVT prevalence data used for this figure). Using aggregated survey data, VT and NVT carriage prevalence were, respectively, 8.4% (95% confidence interval (CI) 5.7–11.8) and 33.8% (28.8–39.1) among children 4–8 weeks (prior to first dose PCV; surveys 5–7); 17.1% (95% CI 14.2–20.5) and 62.7% (58.6–66.6) among PCV-vaccinated children 18 weeks to 1 year old (surveys 4–7); 18.4% (95% CI 15.3–22.1) and 58.5% (54.1–62.8) among PCV-vaccinated children 2 years old (surveys 4–7); 18.0% (95% CI 16.5–19.5) and 56.2% (54.2–58.1) among PCV-vaccinated children 3–7 years old (surveys 1–7); 18.2% (95% CI 16.2–20.3) and 38.5% (35.9–41.0) among PCV-unvaccinated children 3–10 years old (surveys 1–7); and 12.3% (95% CI 10.8–13.9) and 28.1% (26.0–30.2) among HIV-infected adults on ART (surveys 1–7).

A sensitivity analysis among PCV-vaccinated age groups showed that neither the overall VT prevalence nor the VT distribution changed significantly when limiting these analyses to children (i) who received only one, only two, or all three doses PCV; (ii) with document-confirmed PCV vaccination, or (iii) who adhered to the vaccination schedule to within 2 weeks of each scheduled dose. As shown in Supplementary Fig. 1, when stratified by age (in years) and aggregating survey data, reduction in VT carriage was not exponential among vaccinated children. VT carriage increased slightly during the first 4 years of life, from 16.6% (95% CI 11.9–22.6) among children 18 weeks to 11 months old to 17.4% (13.6–21.7), 18.6% (15.1–22.1), and 19.5% (17.3–22.1) among 1-, 2-, and 3-year-old children, respectively. VT carriage then decreased to 18.5% (16.0–20.9), 14.4% (10.8–19.0), 12.0% (6.7–19.3), and 7.0% (1.4–17.1) among 4-, 5-, 6-, and 7 years old, respectively.

Although all 13 VTs were identified in each of the three older (3–40 years old) study groups, with serotype 3 the predominant VT in each, serotype carriage dynamics were more heterogenous among those <3 years old (Fig. 3). Serotype 1, a common cause of IPD in Africa[45,46], contributed 3.0% to the all-ages VT carriage prevalence. Supplementary Tables 7 and 8 show the proportion of total VT carriage attributed to individual VTs, stratified by study group and survey among PCV-vaccinated and PCV-unvaccinated study groups, respectively.

**Table 1 Demographic and household characteristics of study participants.**

| | 4–8 wks old PCV-unvacc'd (n = 346) | 18 wks–1 yr old PCV-vacc'd (n = 566) | 2 yrs old PCV-vacc'd (n = 499) | 3–7 yrs old PCV-vacc'd (n = 2565) | 3–10 yrs old PCV-unvacc'd (n = 1402) | 18–40 yrs old HIV-infected on ART PCV-unvacc'd (n = 1770) |
|---|---|---|---|---|---|---|
| **Demographics** | | | | | | |
| Age yrs, median, (SD) [range] | 0.13 (0.018) [0.08-0.17] | 1.19 (0.46) [0.35-1.99] | 2.50 (0.28) [2.0-2.99] | 4.14 (0.95) [3.0-7.9] | 8.49 (1.63) [3.6-10.99] | 33.55 (5.83) [18.0-40.9] |
| Gender, male n (%) | 179 (51.7) | 296 (52.3) | 244 (48.9) | 1271 (49.6) | 718 (51.2) | 559 (31.96)[c] |
| **PCV received n (%)[a,b]** | | | | | | |
| 1 dose only | — | 1 (02) | 0 | 26 (1.2) | — | — |
| 2 doses only | | 7 (1.2) | 1 (0.2) | 19 (0.9) | | |
| 3 doses | | 558 (98.6) | 498 (99.8) | 2051 (97.9) | | |
| **Household crowding[d]** | | | | | | |
| Crowding index, mean (median)[e] | 2.5 (2.3) | 2.6 (2.5) | 2.5 (2.3) | 2.6 (2.5) | 2.9 (2.5) | 2.1 (2.0) |
| **Children < 5 yrs in HH[e]** | | | | | | |
| Median [range] | 1 [1-3] | 1 [1-3] | 1 [1-2] | 1 [1-3] | 1 [1-3] | 0 [0-4] |
| **Smoker in household[f,g]** | | | | | | |
| Yes, n (%) | 28 (8.1) | 41 (7.2) | 43 (8.6) | 124 (7.7)[g] | 61/674 (9.1) | 33/1092 (3.0) |
| **House structure[f], n (%)** | | | | | | |
| **Walls[e]** | | | | | | |
| Burnt brick/concrete | 174 (50.4) | 168 (29.6) | 169 (33.9) | 941 (36.7) | 901 (64.3) | 1212 (68.5) |
| Unburnt brick | 166 (48.1) | 397 (70.2) | 330 (66.1) | 1621 (63.2) | 488 (34.8) | 292 (16.5) |
| Mud, thick/thin | 6 (1.5) | 1 (0.2) | 0 | 3 (0.10) | 13 (0.9) | 266 (15.0) |
| **Floor** | | | | | | |
| Tiles | 1 (0.3) | 1 (0.2) | 0 | 3 (0.1) | 4 (0.3) | 20 (1.1) |
| Concrete | 325 (93.9) | 447 (89.9) | 406 (89.2) | 2125 (87.3) | 1257 (91.9) | 1619 (91.5) |
| Mud | 20 (5.8) | 48 (9.9) | 49 (10.8) | 307 (12.6) | 107 (7.8) | 130 (7.4) |
| **Latrine[e]** | | | | | | |
| Water toilet | 16 (4.3) | 13 (2.6) | 11 (2.4) | 57 (2.4) | 238 (17.4) | 279 (15.8) |
| Simple pit latrine | 330 (95.7) | 480 (97.0) | 441 (97.6) | 2368 (97.5) | 1126 (82.4) | 2 (0.1) |
| Other | 0 | 2 (0.4) | 0 | 2 (0.1) | 3 (0.2) | 1484 (84.1) |
| **Water[e]** | | | | | | |
| Tap to house | 48 (13.9) | 42 (8.5) | 39 (8.6) | 242 (9.9) | 425 (31.1) | 591 (33.4) |
| Communal tap | 293 (84.9) | 448 (90.1) | 414 (91.0) | 2156 (88.6) | 884 (64.6) | 947 (53.5) |
| Bore hole | 3 (0.9) | 7 (1.4) | 2 (0.4) | 30 (1.2) | 45 (3.3) | 181 (10.2) |
| Well (covered/open) | 2 (0.3) | 0 | 0 | 7 (0.3) | 14 (1.0) | 50 (2.8) |
| **Electricity at household** | | | | | | |
| Yes | 274 (79.4) | 379 (76.3) | 342 (75.2) | 1742 (71.5) | 1021 (74.6) | 1275 (72.1) |
| Possessions index[h] mean (SD)[e] | 7.1 (2.7) | 6.4 (3.5) | 6.4 (3.4) | 6.8 (3.3) | 8.2 (3.2) | 8.2 (3.3) |

Recruitment location: Infants 4–8 weeks were recruited from household or health centre (present for EPI); PCV-vaccinated children 18 weeks to 5 years old were recruited from household; PCV-unvaccinated children 5–10 years old were recruited from schools; HIV-infected adults on ART were recruited from Queen Elizabeth Central Hospital ART Clinic.
PCV pneumococcal conjugate vaccine, ART antiretroviral therapy, SD standard deviation, wk week, yr year, vacc'd vaccinated, HH household.
[a]Among subset of PCV-vaccinated children with written evidence of vaccination.
[b]n = 2096 among children 3–7 yrs old.
[c]The gender distribution among HIV-infected adults recruited from ART Clinic is representative of the gender distribution among those attending the clinic.
[d]Crowding index: Calculated as number of household residents divided by number of bedrooms in main house; data only collected starting survey four.
[e]Distribution of these covariates was statistically significant when comparing unvaccinated children 3–7 years of age and vaccinated children 3–7 years of age. Crowding index (mean 2.9 vs 2.6, respectively) $p < 0.000$; number of children < 5 yrs in household (mean 0.7–1.1) $p < 0.000$; Walls of burnt bricks or concrete (61.5 vs 36.7) $p < 0.000$; Latrine, water toilet (17.4 vs 2.4) $p < 0.000$; Water tap to house (31.1 vs 9.5) $p < 0.000$; Possession index (7.8 vs 6.8) $p < 0.000$.
[f]Smoker in household: the percentage of households with at least one household member who smokes tobacco; data only collected starting survey four. Ranking of household structure variables: each variable is presented with the most costly category at top and least costly at bottom of list.
[g]n = 1615 among children 3–7 yrs old.
[h]Possession index: calculated as a sum of positive responses for household ownership of each of one of fifteen different functioning items: watch, radio, bank account, iron (charcoal), sewing machine (electric), mobile phone, CD player, fan (electric), bednet, mattress, bed, bicycle, motorcycle, car, and television.

**Change in carriage prevalence over time.** To minimise confounding, we calculated relative change in carriage prevalence and prevalence ratios on a narrower age range within each study group (Table 2). Among children 3–5 years old (PCV vaccinated), aggregated (surveys 1–7) VT and NVT carriage prevalence were, respectively, 18.5% (95% CI 16.9–20.1) and 57.2% (55.2–59.2). There was a 16.1% relative reduction in VT carriage, from 19.9% (15.7–25.0) in survey 1 to 16.7% (13.0–21.1) in survey 7. There was a 2.0% relative decrease in NVT carriage, from 64.3%

(58.6–69.7) to 63.0% (57.6–68.1). When adjusted for age (years old) at recruitment, the adjusted prevalence ratio (aPR) over the 3.5-year study was 0.919 (0.845–0.999) $p = 0.047$ for VT carriage and 0.978 (0.944–1.013) $p = 0.208$ for NVT carriage. Among children 6–8 years old (PCV-unvaccinated), aggregated (surveys 1–7) VT and NVT carriage prevalence were, respectively, 18.4% (95% CI 15.6–21.5) and 39.6% (35.9–43.3). There was a 40.5% relative reduction in VT carriage, from 26.4% (18.3–36.4) in survey 1 to 15.7% (8.9–26.3) in survey 7. There was a 12.3%

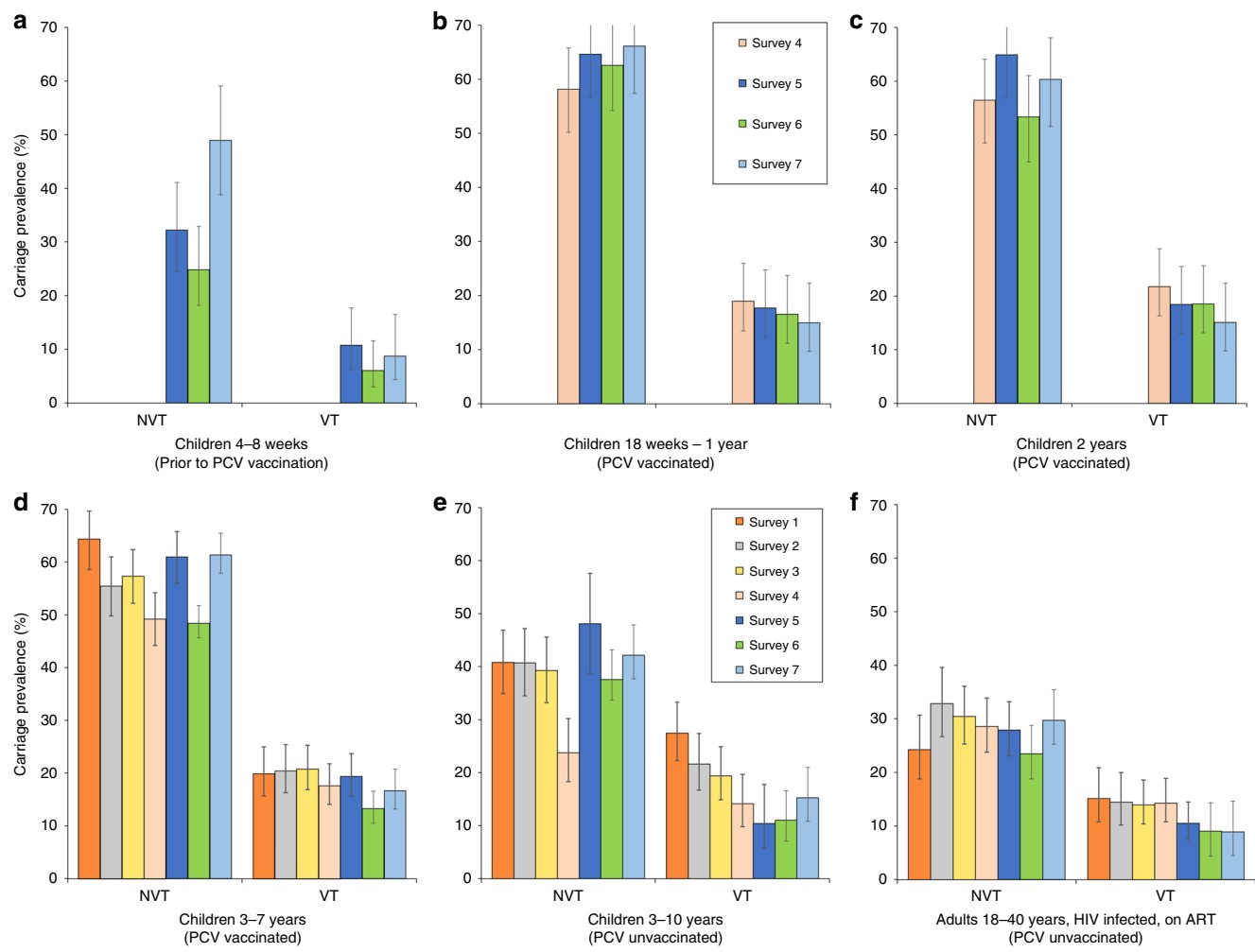

**Fig. 2 *S. pneumoniae* carriage prevalence per survey, stratified by study group.** Surveys 1–7 spanned a time of 3.6–7.1 years after the 12 November 2011 introduction of PCV into Malawi's EPI program. Younger children (4-8 weeks up to 2 years of age; cells **a**–**c**) were recruited starting survey 4 or 5. Prevalence of non-carriers is calculated by 1 − (NVT + VT). Aggregated sample size for each study group: **a** $n = 346$ children 4-8 weeks old (PCV-unvaccinated), **b** $n = 566$ children 18 weeks to 1 year old (PCV vaccinated), **c** $n = 499$ children 2 years old (PCV vaccinated), **d** $n = 2565$ children 3-7 years old (PCV vaccinated), **e** $n = 1402$ children 3-10 years old (PCV-unvaccinated), **f** $n = 1770$ HIV-infected adults on ART (PCV-unvaccinated). Refer to Supplementary Tables 3 and 4 for the sample sizes used in calculating per-survey VT and NVT prevalence data and error bars in this figure. 95% confidence interval error bars are shown. The confidence interval bounds are calculated by exponentiating the bounds in the logit scale.

relative increase in NVT carriage, from 40.7% (31.0–51.1) to 45.7% (34.4–57.5). When adjusted for age at recruitment, the aPR over the 3.5-year study was 0.919 (0.845–0.999) $p = 0.047$ for VT carriage and 0.978 (0.944–1.013) $p = 0.208$ for NVT carriage. Among HIV-infected adults on ART (PCV-unvaccinated), aggregated (surveys 1–7) VT and NVT carriage prevalence were, respectively, 12.3% (95% CI 10.8–13.9) and 28.1% (26.0–30.2). There was a 41.4% relative reduction in VT carriage, from 15.2% (10.8–20.9) in survey 1 to 8.9% (5.7–13.7) in survey 7. There was a 22.7% relative increase in NVT carriage, from 24.2% (18.8–30.7) to 29.7% (23.8–36.4). When adjusted for age at recruitment, the aPR over the 3.5-year study was 0.831 (0.735–0.938) $p = 0.003$ for VT carriage and 0.963 (0.895–1.036) $p = 0.307$ for NVT carriage. To further assess for potential confounders, we investigated whether the demographic characteristics, as reported in Table 1, have changed over the study period. Although there was change in some covariates (Crowding index, Smoker in household, Latrine type, Electricity at household, and Possessions index), the magnitude was limited and, when adjusted for in the model, they made no meaningful difference to the relationship between VT carriage and time.

**Contribution of serotype 3 to change in carriage prevalence.** Several post-PCV13 introduction studies have reported that PCV13 is less immunogenic for serotype 3 compared with other VTs[47,48]. We have therefore assessed this potential bias by classifying serotype 3 as an NVT in a separate analysis (Supplementary Table 9). The aPRs for VT carriage prevalence among PCV-vaccinated children 3–5 years of age followed the same trend but was no longer statistically significant (aPR 0.919 [95% CI 0.845, 0.999] $p = 0.047$ for analysis including serotype 3 as VT vs aPR 0.942 [95% CI 0.856, 1.036] $p = 0.218$ for analysis including serotype 3 as NVT).

**Probability of VT carriage with age and carriage half-life.** Using non-linear regression analysis, with carriage data censored below 3.6 years of age, to investigate the individual probability of VT carriage as a function of a child's age (years), the probability of VT carriage declined with age for both vaccinated and unvaccinated children (Fig. 4). However, the population-averaged effect of not receiving PCV more than doubled an individual's probability of VT carriage at 3.6 years of age, $\beta = 2.15$ (95%

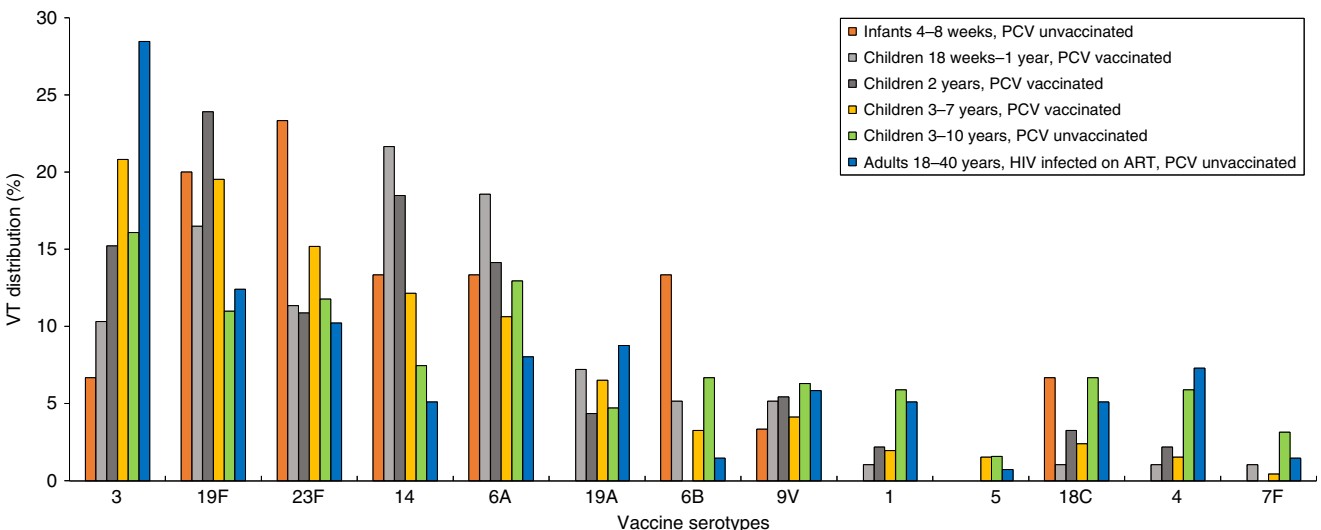

**Fig. 3 Distribution of vaccine-serotype (VT) carriage, aggregated across study period and stratified by study group.** Proportion of VT carriage attributed to individual VTs across all surveys, stratified by study group. The denominator for each serotype is the total VT isolates in each study group.

CI 1.47–2.83) (Fig. 4 and Table 3). While reporting different estimated probabilities of VT carriage at the individual level (e.g. 0.22 for PCV-vaccinated and 0.47 [α*β] for PCV-unvaccinated children at 3.6 years of age), the estimated population-level half-life (derived from individual carriage probability data) of VT carriage prevalence was similar ($T_{1/2}$: 3.34 years [1.78–6.26] vs $T_{1/2}$: 3.26 years [2.42–4.38], respectively). Investigating probability of NVT carriage, β was 0.91 (0.73–1.09), with similar estimated probabilities of NVT carriage for individual vaccinated and unvaccinated children at 3.6 years of age (0.59 and 0.54, respectively [α*β]). The estimated half-life of NVT carriage prevalence was also similar among PCV-vaccinated children ($T_{1/2}$: 9.46 years [4.69–19.04]) and PCV-unvaccinated ($T_{1/2}$: 9.83 years [5.69–16.99]). Assessment of the goodness-of-fit indicates a good fit, with no discernible relationship between the residual and the predicted values and the range of residuals compatible with the theoretical mean and the standard deviations of 0 and 1, respectively (Supplementary Fig. 3).

## Discussion

In this population-based assessment of pneumococcal carriage we surveyed potential reservoir populations from 3.6 to 7.1 years after the introduction of routine PCV13 in Malawi. At the start of the study, we found high VT carriage in all age groups, including younger PCV-vaccinated children, older PCV-unvaccinated children and HIV-infected adults on ART. Despite a statistically significant relative reduction in VT prevalence over the next 3.5 years, residual carriage is higher than reported in high-income settings (Supplementary Table 2). All 13 VTs were isolated despite high vaccine uptake and good schedule adherence. There was no significant change in NVT carriage prevalence. The 18.7% residual VT carriage prevalence we report among PCV-vaccinated children 1–4 years old was consistent with the 16.5% reported in northern Malawi by Heinsbroek et al.[13] for 2014 (Supplementary Table 5). Though the residual VT carriage prevalence we report for Blantyre was lower than that observed by Heinsbroek et al. before vaccine introduction (28%)[13], and may have missed the greatest fall in VT prevalence soon after PCV introduction, we did not observe the substantially lower levels rapidly achieved in high-income low carriage prevalence settings (<5%) associated with control of carriage and transmission[17,26,27]. When aggregating surveys, reduction in VT carriage among PCV-vaccinated children did not conform to a simple exponential distribution, with a non-statistically significant increase among children under 4 years of age, before evidence of a decrease among older vaccinated children (Supplementary Fig. 1). In the light of the recent WHO Technical Expert Consultation Report on Optimisation of PCV Impact[49], these data start to address the paucity of information on the long-term impact of the widely implemented 3 + 0 vaccine schedules on serotype-specific disease and carriage in this region.

Despite the reporting of high residual VT carriage, our non-linear statistical analysis shows a lower probability of VT carriage among vaccinated children, starting at 3.6 years of age when the model is censored. From this we speculate that vaccine-induced protection, despite evidence of early waning of immunity (perhaps within the first 6–12 months of life)[42,50], does provide a longer-term benefit to this population in providing a lower VT carriage setpoint at an age when naturally acquired and herd immunity begins to play a role in carriage control. A lower carriage setpoint among vaccinated children is of benefit in both direct protection and in reducing transmission. We propose that a comparable VT half-life at 3.6 years of age is due to a limited role of vaccine-induced immunity that started to wane within the first 12 months of life. The mechanism underlying the vaccine effect could be prevention of carriage (reduced incidence) or shortening of carriage duration (reduced point prevalence). There is also evidence that, in older vaccinated and unvaccinated children, the reductions in carriage prevalence are due to the indirect benefits of vaccination augmented by naturally acquired immunity to subcapsular protein antigens[51,52].

These indirect benefits augmented by naturally acquired immunity explains, in part, the more pronounced decline in VT carriage among PCV-unvaccinated children 6–8 years old (40.5%) and HIV-infected adults on ART (41.4%), compared with younger vaccinated children 3–5 years old (16.1%; Table 2). Using a dynamic transmission model fitted to data from this population in Blantyre, Malawi, we have shown that the force of infection (FOI; the rate by which a certain age group of susceptible individuals is infected) is characterised by different transmission potentials within and between age groups[53]. This analysis suggests that the time period of the fastest FOI reduction for the 0–5 years old was between vaccine introduction and 2015 (when no carriage data were collected), which contrasted with the older age groups, for which the period of the fastest FOI reduction was predicted to be just before or during the first three surveys.

**Table 2 Carriage prevalence among children and adults.**

| | Children 3-5 years PCV-vaccinated % (n); 95% CI | Children 6-8 years PCV-unvaccinated % (n); 95% CI | Adult, HIV-infected on ART PCV-unvaccinated % (n); 95% CI |
|---|---|---|---|
| Survey 1 | n = 286 | n = 91 | n = 198 |
| Total carriage | 84.2 (241) 79.5, 88.3 | 67.1 (61) 56.4, 76.5 | 39.4 (78) |
| VT | 19.9 (57); 15,7, 25.0 | 26.4 (24) 18.3, 36.4 | 15.2 (30) 10.8–20.9 |
| NVT | 64.3 (184); 58.6, 69.7 | 40.7 (37) 31.0, 51.1 | 24.2 (48) 18.8–30.7 |
| Survey 2 | n = 303 | n = 111 | n = 201 |
| Total carriage | 76.0 (230) 70.7, 80.6 | 65.7 (73) 56.2, 74.5 | 47.2 (95) |
| VT | 20.5 (62) 16.3, 25.4 | 21.6 (24) 14.9, 30.3 | 14.4 (29) 10.2–20.0 |
| NVT | 55.5 (168) 49.8, 61.0 | 44.1 (49) 35.2, 53.5 | 32.8 (66) 26.7–39.6 |
| Survey 3 | n = 361 | n = 139 | n = 279 |
| Total carriage | 78.1 (282) 73.5, 82.3 | 63.3 (88) 54.7, 71.3 | 44.5 (124) |
| VT | 20.8 (75) 16.9, 25.3 | 21.6 (30) 15.5, 29.2 | 14.0 (39) 10.4–18.6 |
| NVT | 57.3 (207) 52.2, 62.4 | 41.7 (58) 33.8, 50.1 | 30.5 (85) 25.3–36.1 |
| Survey 4 | n = 378 | n = 128 | n = 308 |
| Total carriage | 67.0 (253) 61.9, 71.7 | 42.9 (55) 34.3, 52.0 | 42.9 (132) |
| VT | 17.5 (66) 14.0, 21.6 | 14.8 (19) 9.6, 22.2 | 14.3 (44) 10.8–18.9 |
| NVT | 49.5 (187) 44.4, 54.5 | 28.1 (36) 21.0, 36.6 | 28.6 (88) 23.8–33.9 |
| Survey 5 | n = 371 | n = 56 | n = 305 |
| Total carriage | 80.6 (299) 76.2, 84.5 | 55.3 (31) 41.5, 68.7 | 38.4 (117) |
| VT | 19.4 (72) 15.7, 23.8 | 8.9 (5) 3.7, 19.9 | 10.5 (32) 7.5–14.5 |
| NVT | 61.2 (227) 56.1, 66.0 | 46.4 (26) 33.8, 59.6 | 27.9 (85) 23.1–33.2 |
| Survey 6 | n = 382 | n = 100 | n = 277 |
| Total carriage | 67.3 (257) 62.3, 72.0 | 52.0 (52) 41.8, 62.1 | 32.5 (90) |
| VT | 15.2 (58) 11.9, 19.2 | 15.0 (15) 9.2, 23.5 | 9.0 (25) 6.2–13.0 |
| NVT | 52.1 (199) 47.1, 57.1 | 37.0 (37) 28.1, 46.9 | 23.5 (65) 18.8–28.8 |
| Survey 7 | n = 324 | n = 70 | n = 202 |
| Total carriage | 79.7 (258) 74.8, 83.9 | 61.4 (43) 49.0, 72.8 | 38.6 (78) |
| VT | 16.7 (54) 13.0, 21.1 | 15.7 (11) 8.9, 26.3 | 8.9 (18) 5.7–13.7 |
| NVT | 63.0 (204) 57.6, 68.1 | 45.7 (32) 34.4, 57.5 | 29.7 (60) 23.8–36.4 |
| Total (Survey 1-7) | n = 2405 | n = 695 | n = 1770 |
| Total carriage | 75.7 (1820) 73.9, 77.4 | 58.0 (403) 54.2, 61.7 | 40.4 (714) |
| VT | 18.5 (444) 16.9, .20.1 | 18.4 (128) 15.6, 21.5 | 12.3 (217) 10.8–13.9 |
| NVT | 57.2 (1376) 55.2, 59.2 | 39.6 (275) 35.9, 43.3 | 28.1 (497) 26.0–30.2 |
| | cPR[a] (95% CI) p-value | cPR[a] (95% CI) p-value | cPR[a] (95% CI) p-value |
| Total carriage | — | — | — |
| VT | 0.912 (0.840, 0.990) 0.028 | 0.806 (0.690, 0.942) 0.007 | 0.847 (0.750, 0.957) 0.008 |
| NVT | 0.972 (0.939, 1.007) 0.112 | 0.964 (0.881, 1.056) 0.437 | 0.967 (0.899, 1.040) 0.362 |
| | aPR[a] (95% CI) p-value | aPR[a] (95% CI) p-value | aPR[a] (95% CI) p-value |
| Total carriage | — | — | — |
| VT | 0.912 (0.840, 0.990) 0.028 | 0.839 (0.712, 0.990) 0.037 | 0.831 (0.735, 0.938) 0.003 |
| NVT | 0.972 (0.939, 1.007) 0.112 | 0.974 (0.886 1.070) 0.583 | 0.963 (0.895, 1.036) 0.307 |
| | Relative change[b] | Relative change[b] | Relative change[b] |
| Total carriage | — | — | — |
| VT | −16.1% | −40.5% | −41.4% |
| NVT | −2.0% | +12.3% | 22.7% |

Surveys 1–7 spanned a time of 3.6–7.1 years after Malawi's November 2011 introduction of PCV.
cPR crude prevalence ratio, aPR adjusted prevalence ratio (adjusted for age [years old] at recruitment), CI confidence interval, VT vaccine serotype, NVT non-vaccine serotype, ART antiretroviral therapy.
[a]Carriage prevalence ratios (crude and adjusted) were calculated over the study duration of seven surveys by log-binomial regression using years (365.25 days) between date of Malawi's PCV introduction and participant recruitment, coded as a single time variable, allowing an estimate of (adjusted) prevalence ratio per annum.
[b]Relative change = [(VT prevalence of final survey − VT prevalence of initial survey)/VT prevalence of initial survey] × 100%.

To achieve herd protection in settings with high carriage prevalence, such as Malawi, we need to effectively interrupt person-to-person transmission. In Finland, a microsimulation model suggested a moderate transmission potential of pneumococcal carriage, predicting the elimination of VT carriage among those vaccinated within 5–10 years of PCV introduction, assuming high (90%) vaccine coverage and moderate (50%) vaccine efficacy against acquisition[54]. Thus, vaccine impact predicted by transmission models from low carriage prevalence settings probably does not translate to high carriage prevalence settings. Although it has previously been assumed that PCVs would eliminate VT carriage in mature PCV programmes[55], our data bring into

question the potential for either a sustained direct or indirect effect on carriage using a 3 + 0 strategy. Alternative vaccine schedules including those with a booster, should be evaluated to determine whether a higher rate of vaccine-induced VT carriage decay can be achieved.

In Malawi, the vaccine impact on carriage prevalence has been less than that observed in Kenya, The Gambia and South Africa which have used different vaccination strategies. Kenya reported a reduction from 34 to 9% VT carriage among PCV-vaccinated children under 5 years of age, 6 years after introduction of 10-valent PCV[22]. The Gambia reported a reduction from 50% to 13% VT carriage among children 2–5 years old, 20 months after

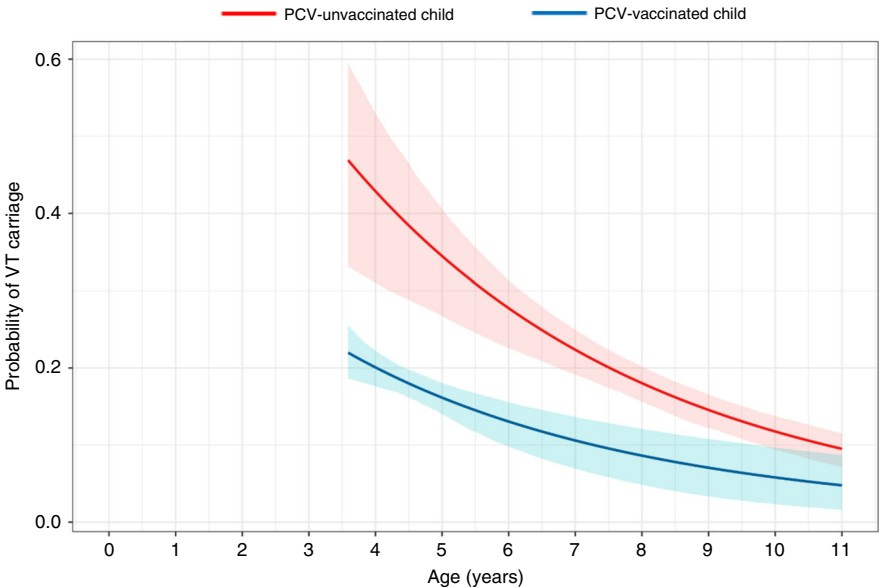

**Fig. 4 Modelling the relationship between a child's probability of VT carriage and age.** Estimated probabilities and pointwise 95% confidence intervals (shaded regions) of the probability of an individual child's vaccine-serotype (VT) carriage as a function of a child's age (years), for an unvaccinated child (red line) and a vaccinated child (blue line). The fitted line for unvaccinated children includes the range of the empiric data. The fitted line for vaccinated children is left censored at 3.6 years old and extrapolated beyond the oldest vaccinated child (7.9 years old). The model shows significantly different estimated probabilities of VT carriage for an individual (distance between lines), while the estimated population-level half-life of VT carriage (derived from individual carriage probability data; refer to Table 3 in manuscript) translates to very similar estimates among PCV-vaccinated (3.34 years) and PCV-unvaccinated (3.26 years) children.

**Table 3 Estimates for probability of carriage with age and carriage half-life.**

| Parameter | VT | | NVT | |
|---|---|---|---|---|
| | Estimate | 95% CI | Estimate | 95% CI |
| Carriage prevalence at censoring age (3.6 years) for vaccinated children ($\alpha$) | 0.22 | 0.19, 0.25 | 0.59 | 0.55, 0.63 |
| Carriage prevalence at censoring age (3.6 years) for unvaccinated children ($\alpha$*$\beta$) | 0.47 | 0.33, 0.60 | 0.54 | 0.43, 0.64 |
| Decay rate of carriage prevalence with age for vaccinated children ($\delta_v$) | 0.21 | 0.11, 0.39 | 0.07 | 0.04, 0.15 |
| Carriage half-life for vaccinated children (log (2)/$\delta_v$) | 3.34 | 1.78, 6.26 | 9.46 | 4.69, 19.04 |
| Decay rate of carriage prevalence with age for unvaccinated children ($\delta_u$) | 0.21 | 0.16, 0.29 | 0.07 | 0.04, 0.12 |
| Carriage half-life for unvaccinated children (log (2)/$\delta_u$) | 3.26 | 2.42, 4.38 | 9.83 | 5.69, 16.99 |
| Effect of not receiving PCV ($\beta$) | 2.15 | 1.47, 2.83 | 0.91 | 0.73, 1.09 |

introducing the 7-valent PCV[56]. Likewise, a study from South Africa showed reduced PCV13-serotype colonisation from 37% to 13% within 1 year of transitioning from PCV7 to PCV13[57]. However, these countries have also not achieved the low carriage prevalence seen in Europe and North America 2–3 years post introduction[4,18]. As presented by Lourenço et al.[53], we propose that a high FOI in settings such as Malawi limits a 3 + 0 schedule to achieving only a short duration of VT carriage control in infants. While a 2 + 1 schedule, as deployed in South Africa, may improve colonisation control, this remains unproven in other African settings. Given the likely importance of an early reduction in transmission intensity to maintain a reduced carriage prevalence, a catch-up-campaign with booster doses over a broader age range (i.e. <5 years of age) may also be required. Although GAVI has considerably reduced PCV costs for low-income countries[58,59], vaccine impact must be optimised (particularly indirect effects) to achieve financial sustainability. The FOI and determinants of transmission between and within age groups need to be considered, as new approaches to improving vaccine-induced carriage reduction are proposed and tested.

Unlike low-transmission settings[60], as well as The Gambia[29] and South Africa[57], we observed a very modest decrease in NVT carriage among young PCV-vaccinated children in Malawi. Given evidence elsewhere of rapid serotype replacement after PCV introduction, it is possible that serotype replacement and redistribution had already occurred before the start of this study, and that as part of a stochastic secular trend, we are now observing an overall decrease in pneumococcal carriage prevalence. There may have also been individual NVTs that increased, while other NVTs decreased in prevalence. Though distribution of individual NVTs warrant further analysis, our latex serotyping methods did not allow for identifying individual NVT serotypes. It is also plausible that overall improvement in living conditions (improved nutrition, sanitation, and disease control) and health care (anti-retroviral roll-out and rotavirus vaccination) have resulted in an overall sustained drop in pneumococcal carriage as a result of improved health, evidenced by falling under 5 mortality in recent years[61]. Either way, the importance of this in NVT carriage will become clearer as the trends in NVT invasive disease become available from these different settings.

We have previously shown incomplete pneumococcal protein antigen-specific reconstitution of natural immunity and high levels of pneumococcal colonisation in HIV-infected Malawian adults on ART[43,44]. While reporting here a significant decline

(41.4% relative reduction, Table 2) in VT carriage among an HIV-infected adult population suggests some indirect benefit in this population following routine infant PVC13 introduction, the residual 8.9% VT carriage may represent a persistent reservoir of VT carriage and transmission. Previous studies in Malawi and South Africa have suggested that despite a higher risk of VT pneumococcal colonisation among HIV-infected women, they are still unlikely to be a significant source of transmission to their children[24]. However, in the context of routine infant PCV13 and persistent pneumococcal carriage, the balance of transmission may now be different. Given the higher risk of IPD, ongoing burden of pneumococcal pneumonia[62], and the evidence that PCV protects HIV-infected adults from recurrent VT pneumococcal infections[63], targeted vaccination benefitting this at-risk population may help reduce overall carriage and disease prevalence.

Though this work provides a robust community-based estimate of VT and NVT pneumococcal carriage in Blantyre, there are some limitations worth noting. The study was conducted over a relatively short timeframe for understanding long-term temporal trends. For this reason, the statistical analysis has some limitations in its ability to disentangle the effects of calendar time and age since vaccination, given the small overlap in ages of vaccinated and unvaccinated children in our data (Supplementary Fig. 2). It is also possible that re-adjustment of carriage dynamics (VT and NVT prevalence as well as serotype-specific trends) have already occurred between PCV introduction and our first carriage survey. As reported elsewhere[64,65], a major challenge in field-based surveillance studies is ascertainment of vaccine coverage. Measurement of vaccination status depends predominantly on health passport with limited capacity for verification of either recorded or reported vaccination status. However, reported high coverage is concordant to that reported by other studies in Malawi and any misclassification is therefore likely to be small and would not significantly change the findings. Jahn et al.[65] showed that Bacillus Calmette–Guerin (BCG) scar data allow inference of population vaccination coverage independently from vaccination records, reporting a similar prevalence of BCG scar among children < 5 years with no health passport compared with those with a health passport and BCG reported (70% vs 78%, respectively). Subsequently, a 2015 Malawi cluster vaccination coverage survey reported that 94% of children 12–23 months of age had a BCG scar[36]. Although there are pre-vaccine-introduction data from elsewhere in Malawi, there are no equivalent historical carriage data for urban settings in Malawi using the same sampling frame. However, this does not detract from the finding of high levels of residual VT carriage in these reservoir populations. Finally, given evidence that more sensitive serotyping methods that detect multiple serotype carriage (e.g. by DNA microarray)[66,67] will increase VT carriage estimates, our carriage prevalence data likely underestimate the true residual VT, as well NVT, prevalence levels.

In conclusion, despite success in achieving direct protection of infants against disease, a 3 + 0 PCV13 schedule in Malawi has not achieved the low universal VT carriage prevalence reported in high-income settings and that is required to control carriage and transmission. We propose that although vaccine-induced immunity reduces the risk of VT carriage in children, in the context of a high residual FOI, this impact is limited by rapid waning (perhaps within the first 6-12 months of life) of vaccine-induced mucosal immunity and pneumococcal recolonisation. Therefore, alternative schedules and vaccine introduction approaches in high pneumococcal carriage, high-disease-burden countries should be revisited through robust evaluation rather than through programmatic change without supporting evidence. Furthermore, we need to better understand the relative impact of waning vaccine-induced immunity, indirect vaccine protection and naturally acquired immunity on VT carriage in 2–3 years after vaccination.

## Methods

**Study design.** This was a prospective observational study using stratified random sampling to measure pneumococcal nasopharyngeal carriage in Blantyre, Malawi. Sampling consisted of twice-annual rolling cross-sectional surveys over 3.5 years.

**Study population.** Blantyre is located in southern Malawi and has an urban population of ~1.3 million. Recruitment included four groups: (i) healthy infants 4–8 weeks old prior to the first dose of PCV, recruited from vaccination centres using systematic sampling; (ii) randomly sampled healthy children 18 weeks to 7 years old who received PCV as part of routine EPI or via the catch-up campaign, recruited from households and public schools; (iii) randomly sampled healthy children 3–10 years old who were age ineligible (born on or before 11 November 2010 and, therefore, too old to receive PCV as part of routine EPI or via the catch-up campaign), recruited from households and public schools; and (iv) HIV-infected adults 18–40 years old and on ART, recruited from Blantyre's Queen Elizabeth Central Hospital ART Clinic using systematic sampling. After evidence of persistent carriage among children 3–10 years old during the early surveys, recruitment of infants 4–8 weeks old and children 18 weeks to 2 years old was implemented starting from survey 5 and survey 4, respectively. Exclusion criteria for all participants included current tuberculosis treatment, pneumonia hospitalisation ≤14 days prior to screening, or terminal illness. Exclusion criteria for children included parent/guardian not providing written informed consent, child 8–10 years old not providing written informed assent, reported immunocompromising illness (including HIV), having received antibiotics ≤14 days prior to screening, having received PCV if age ineligible or not having received PCV if age eligible. Individuals were not purposely resampled but were eligible if randomly re-selected in subsequent surveys.

**Site selection and recruitment.** Households, schools, and vaccination centres were selected from within three non-administrative zones representative of urban Blantyre's socioeconomic spectrum in medium- to high-density townships. These zones were further divided into clusters, allowing for ~25,000 adults per zone and 1200 adults per cluster. At the start of each rolling cross-sectional survey, eight clusters were randomly selected. Clusters were not purposely resampled but eligible if randomly selected in subsequent surveys. Within each cluster, after randomly choosing a first house, teams moved systematically, recruiting one eligible child per household until a required number of children were recruited from each cluster. If no parent/guardian was available or no child age eligible for household recruitment was at home (including when no one was home), the study team moved systematically to the next house without attempting to revisit the skipped household. At the start of each survey, updated school registers were collected from schools. Individual school-goers were randomly selected from school registers, and letters were sent home inviting parents/guardians to travel to the school within the following 3 school days to discuss the study and consider consenting to their child's participation. If the parent/guardian did not respond to the letter or did not visit the school within the specified 3 days, another child was randomly selected from the school's register.

**Determining PCV vaccination status.** A child was considered PCV vaccinated if she/he had received at least one dose of PCV before screening. Vaccination status and inclusion/exclusion criteria were further assessed from subject-held medical records (known as health passports). If a child was reported by the parent/guardian to be PCV vaccinated but no health passport was available, a questionnaire was applied. The questionnaire was developed by identifying, among a subset of 60 participants, four questions most commonly answered correctly by parents/guardians of children with proof of PCV vaccination. The questions included the child's age when vaccinated, vaccine administration route (oral or injectable), anatomical site of vaccination, and which other (if any) vaccines were received at the time of PCV vaccination. If the child was PCV age eligible and the parent/guardian answered all four questions correctly, the child was recruited as PCV vaccinated.

**Sample size.** The sample size strategy was pragmatic to allow for adequate precision of the carriage prevalence estimates. VT carriage was considered the primary endpoint, and the sample size was calculated based on the precision of the prevalence estimation, assuming an infinite sampling population. Among children 3–7 years old (vaccinated), an absolute VT prevalence up to 10% was expected, with a sample of 300/survey providing a 95% CI of 6.6–13.4%. Among children 3–10 years old (unvaccinated) and HIV-infected adults, an absolute VT prevalence of 20% was expected, with a sample of 200/survey providing a 95% CI of 14.5–25.5%.

**Nasopharyngeal swab collection.** An NPS sample was collected from each participant using a nylon flocked swab (FLOQSwabs[TM], Copan Diagnostics, Murrieta, CA, USA), and then immediately placed into 1.5 mL skim

milk–tryptone–glucose–glycerol (STGG) medium and processed at the Malawi–Liverpool–Wellcome (MLW) Trust laboratory in Blantyre, according to WHO recommendations[68]. Samples were frozen on the same day at −80 °C.

**Pneumococcal identification and latex serotyping**. After being thawed and vortexed, 30 µL NPS–STGG was plated on gentamicin-sheep blood agar (SBG; 7% sheep blood agar, 5 µL gentamicin/mL) and incubated overnight at 37 °C in 5% $CO_2$. Plates showing no *S. pneumoniae* growth were incubated overnight a second time before being reported as negative. *S. pneumoniae* was identified by colony morphology and optochin disc (Oxoid, Basingstoke, UK) susceptibility. The bile solubility test was used on isolates with no or intermediate (zone diameter < 14 mm) optochin susceptibility. A single colony of confirmed pneumococcus was selected and grown on a new SBG plate as before. Growth from this second plate was used for serotyping by latex agglutination (ImmuLex™ 7-10-13-valent Pneumotest; Statens Serum Institute, Denmark). This kit allows for differential identification of each PCV13 VT but not for differential identification of NVT serotypes; NVT and non-typeable isolates were, therefore, reported as NVT. Samples were batch tested on a weekly basis, blinded to the sample source. Latex serotyping results showed good concordance with whole genome sequence and DNA microarray serotyping[69].

**Statistical analysis**. Participant demographic characteristics were summarised using means, standard deviations, medians, and ranges for continuous variables and frequency distributions for categorical variables. Non-ordinal categorical variables were assessed as indicators. Comparison of covariate distribution between study groups was done by independent group *t*-test for continuous covariates and by chi-square ($\chi^2$) for categorical covariates. Potential confounders were identified by testing the association between variables and VT carriage and included in the multivariable models when $p < 0.1$. Carriage crude prevalence ratios and aPRs were calculated over the study duration by log-binomial regression using years (365.25 days) between Malawi's introduction of PCV13 (12 November 2011) and participant recruitment, coded as a single time variable. This allowed for an estimated prevalence ratio per annum. Prevalence ratio analyses were restricted to children 3–5 years old (PCV vaccinated), children 6–8 years old (PCV-unvaccinated) and adults. This (i) limited bias due to age, (ii) maximised the chance that any change in carriage prevalence over time to be represented primarily by change in calendar time (not ageing in time), and (iii) maximised the inclusion of age category representatives with sequential years of the same PCV vaccination status. The formula for relative change in carriage prevalence was: ([carriage prevalence of final survey − carriage prevalence of initial survey]/carriage prevalence of initial survey) × 100%. Confidence intervals are binomial exact. Statistical significance was inferred from two-sided $p < 0.05$. Participant data collection was completed using the Open Data Kit Collect open-source software. (v1.24.0). Statistical analyses were completed using Stata 13.1 (StataCorp, College Station, TX, USA).

**Non-linear regression analysis for VT carriage half-life**. To better understand the rate at which VT and NVT carriage prevalence was decreasing, we developed a non-linear model to describe the variation in individual probability of VT or NVT carriage with age, adjusted for age at recruitment. The model is fitted using carriage data from children 3.6–10 years of age, maximising overlap with empirical data and allowing direct comparisons of parameters between vaccinated and unvaccinated children. Model outputs for individual VT and NVT carriage probability were then transformed into a population-level (decay) half-life of each VT and NVT carriage (i.e. time in years for VT and NVT carriage prevalence in the sampled cohort to reduce to one-half of its peak), using $\log(2)/\delta$, where $\delta$ = rate of decay of VT or NVT carriage prevalence with age. Model parameters were estimated by maximum likelihood, and 95% confidence bands for the predicted exponential decay curves are obtained through parametric bootstrap. This analysis used R open-source software, version 3.5.0 (www.r-project.org). Details of the analysis framework are included as Supplementary Note 1.

**Consent to participate**. Adult participants and parents/guardians of child participants provided written informed consent, children 8-10 years old provided informed assent. This included consent for publication.

**Reporting summary**. Further information on research design is available in the Nature Research Reporting Summary linked to this article.

## Data availability
The data supporting the findings of this study have been deposited in the Figshare repository (https://doi.org/10.6084/m9.figshare.11985255) (ref. [70]). The source data underlying Figs. 2 and 3 are available in the Supplementary Information, Supplementary Tables 3 and 4, respectively. Figure 4 data are available with the coding data at: https://github.com/claudiofronterre/pneumococco.

## Code availability
The computer coding used to generate results in this manuscript is available at: https://github.com/claudiofronterre/pneumococco.

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

## Acknowledgements

We thank the individuals who participated in this study and the local schools and authorities for their support. We are grateful to the study field teams (supported by Farouck Bonomali and Roseline Nyirenda) and the study laboratory team. We are grateful to the hospitality of the QECH ART Clinic, led by Ken Malisita. Our thanks also extend to the MLW laboratory management team (led by Brigitte Denis) and the MLW data management team (led by Clemens Masesa). R.S.H., N.F., and T.S. are supported by the National Institute for Health Research (NIHR) Global Health Research Unit on Mucosal Pathogens using UK aid from the UK Government. The views expressed in this publication are those of the author(s) and not necessarily those of the NIHR or the Department of Health and Social Care.

## Author contributions

T.D.S, N.B-Z., D.E., N.F., and R.S.H. designed the study. T.D.S, C.F., J.L., U.O., A.G., N.B-Z., D.E., A.W.K., T.S.M., A.A.M., C.M., M.B., S.G., P.D., N.F., and R.S.H. contributed

to the development or design of methodology. T.D.S., N.F. and R.H.S. oversaw the study, data collection, and data management. C.F. and P.D. developed the statistical regression analysis. T.D.S., N.B-Z, N.F., and R.S.H. conducted the statistical analysis. T.D.S., N.F., and R.S.H. wrote the first draft of the paper. T.D.S., C.F., J.L., U.O., A.G., N.B-Z., D.E., A.W.K., T.S.M., A.A.M., C.M., M.B., S.G., P.D., N.F., and R.S.H. contributed to subsequent drafts and read and approved the final version of the report. Bill & Melinda Gates Foundation, USA. A project grant jointly funded by the UK Medical Research Council (MRC) and the UK Department for International Development (DFID) under the MRC/DFID Concordat agreement, also as part of the EDCTP2 programme supported by the European Union (MR/N023129/1); and a recruitment award from the Wellcome Trust (Grant 106846/Z/15/Z). The MLW Clinical Research Programme is supported by a Strategic Award from the Wellcome Trust, UK. National Institute for Health Research (NIHR) Global Health Research Unit on Mucosal Pathogens using UK aid from the UK Government. The views expressed in this publication are those of the author(s) and not necessarily those of the NIHR or the Department of Health and Social Care. The funders had no role in study design, collection, analysis, data interpretation, writing of the report or in the decision to submit the paper for publication. The corresponding author had full access to the study data and, together with the senior authors, had final responsibility for the decision to submit for publication.

## Competing interests

N.B-Z. reports investigator-initiated research grants from GlaxoSmithKline Biologicals and from Takeda Pharmaceuticals outside the submitted work. No other authors declare competing interests.

Ethics approval
The study protocol was approved by the College of Medicine Research and Ethics Committee, University of Malawi (P.02/15/1677) and the Liverpool School of Tropical Medicine Research Ethics Committee (14.056).
