## [Peer Review File · Nature Communications]

Reviewers' Comments:

Reviewer #1:

Remarks to the Author:

This study shows the vaccine-type (VT) carriage remaining after 3-6 years of PCV13 use in an urban area in Malawi. It describes carriage in vaccinated and unvaccinated children and unvaccinated HIV+ adults in a setting with high PCV coverage using a 3+0 schedule under conditions of high pneumococcal carriage. This study adds to the literature on the epidemiology of pneumococcal carriage after long-term use of PCV in low-income settings. However, the paper needs some attention from more senior, experienced writers and pneumococcal experts since the evidence shown is insufficient to support many of the claims made by the authors and inferences on mechanisms of action of immunity, specifically vaccine-induced vs immunity induced by natural exposure, and on duration of vaccine-induced immunity go beyond the limits of the data. The below items include many issues that need to be addressed in this manuscript, but likely not all as I stopped at line 353 when it was clear that more senior guidance in general is required.

Major items to address

1. Results: PCV13 is not effective in preventing ST3 carriage, so also estimate impact counting ST3 as a non-vaccine-preventable ST. This will restrict analyses to the STs that are vaccine-preventable. This eliminates any bias when comparing settings on the contributions of ST3 (i.e.,
2. Abstract conclusions: Line 23 - The wording "compared to HICs, the 3+0 schedule in Malawi has led to a suboptimal reduction in pneumococcal carriage prevalence" (1) goes beyond the scope of the data (no HICs were studied) and (2) implies any differences are due to the schedule, not the setting (higher overall spn carriage, higher prevalence of risk factors, higher HIV prevalence, etc.). Line 24 - The wording "This is likely due to recolonisation of vaccinated children with waning vaccine-induced immunity.." is speculative and not supported by the data. Conclusions should be based only on the evidence in the data; any speculations must be worded as such.
3. Background needs attention from a more experienced investigator. It reads as if it was not thoughtfully composed. Examples of issues include:
 - a. the pneumococcal deaths estimate cited is for the pre-PCV era and two decades ago (2000). Deaths have come down considerably since then. Use more recent (2015) estimates (B. Wahl, Lancet, 2018).
 - b. results of the previous post-PCV carriage study in Malawi are not described, only their pre-PCV results (ref #11). This is important because need to show what has already been done and to justify why another carriage study is needed. (I think it is - because the other study was early post-PCV (yr 2 post), had small N, and adults were HIV- mothers of infants rather than HIV+ adults.)
 - c. "differing markedly from high-income settings" - only cited African LIC data but need to support this statement by also citing HIC data.
 - d. Since 'markedly' is subjective, need to also state the range of VT carriage in HICs to compare to the estimates provide for LICs.
 - e. Add ref 19 to ref list '15-18' on line 55
 - f. The residual carriage estimates cited (refs 22-24, line 57) include only one PCV10/13 using country; the other studies are impact with PCV7. Impact between HICs and LICs should be compared using the same serotypes since (1) PCV10/13 'VT' carriage is higher because it has more STs, and (2) PCV7 was effective in preventing all STs it covered whereas PCV13 does not protect against ST3 carriage (and probably slow-acting if any efficacy against IPD).
 - g. Line 57: the claim of rapid onset of NVT replacement in carriage in the Africa region implies it does not occur everywhere - this is false. NVT replacement is a known phenomenon of spn carriage everywhere.
 - h. Line 58-60: this sentence implies carriage replacement with NVT types means disease replacement. This is only true if they are virulent like the VTs, but they are not equally so. Read the literature on IPD replacement with NVTs in HICs and LICs and revise.

i. Lines 60-62: this sentence about 3+0 schedule is a non-sequitur. There is no mention of other schedule choices and what would be preferred. Or what this schedule has to do with anything discussed so far.

4. The PCV coverage in this community is inadequately described and the description of the validation method for determining vaccination status of older kids (which only 25% had immunization card) is insufficient. Only 60 children were evaluated and the number that were not vaccinated, which is the validation in question, is not described. Nor were any details on the sensitivity and specificity of the findings presented. The level and potential for misclassification of vaccination status is important to understand since this defines the primary analysis comparison groups. The vaccination coverage in infants over the past years in this community is also important since it describes the conditions in which this carriage data represent. Other evidence should be given to describe the annual coverage estimates rather than parent recall of something that happened many years ago. I do not think it is reasonable to assume that parents of a 7 year old remember what legs/arms were jabbed, how many shots, etc., their child received 7 years ago. They may be remembering even the vaccination of a younger sibling, if at all.

5. Since the analysis compares changes in VT carriage over time (Tables 2-3) and by age (Figure 4), describing any differences in the Table 1 characteristics by these factors would show whether any are potential confounders.

6. Describe the units for the prevalence ratio – is this comparing month to month?? It would more informative if shown as annual change (change over 12 months).

7. Line 259: authors state the reduction in VT carriage among vaccinated children was not exponential and cite Suppl 1. Yet I could not find where this decay function was shown to be ruled out. I suspect they did not have the statistical power to rule out this possibility, especially given they are likely at the tail end of the impact since the study started after 3.5 years of PCV use, after when the majority of the decline would be expected.

8. The study is completely confounded by age and year post-PCV intro as there is no meaningful overlap in any survey period of vaccinated and unvaccinated children of the same age. This means effects of age and time cannot be disentangled and so no assumptions about differences in VT carriage can be attributed to either age or time. One suggestion, if there is sufficient sample size, is to show VT carriage in Figure 4 by survey (i.e., multiple red lines and multiple blue lines) and restrict to ages the vaccinated and unvaccinated populations are assessed in most time periods. For vaccinated (blue line) this would be ages 4-5 years and ages 6-7 years for unvac (red). This would eliminate the confounding the other ages contribute to vacc and unvac groups since they differ greatly in their age distribution. Then can see if there is a smooth transition between blue 5 to red 6 years or if there appears to be a jump relative to the changes from blue 4-5 and red 6-7 in any given time period. Regardless, because of this age-time confounding problem, the results and conclusions should not focus in this analysis, but rather on the more interesting long-term VT carriage prevalence observed. The decay with age is well known and has been documented pre-PCV so that analysis adds nothing there. And the rate or shape of the decline has not been made relevant to understand from a policy point of view why we should care about it. Perhaps it could be used to speculate on how long it will take for elimination of VT carriage in this community, but I think that goes beyond the limits of this data.

9. Methods for calculating 'relative change' in Table 2-3 is not described. This should also be presented with CIs.

10. For external comparability with the previous NP carriage study done in Malawi, present carriage results for 1-4 year olds (as opposed to only stratified 18k-1 year and 2 years and 3-7/10 years) and show for <5 year olds (18wk-4y) for external comparability more broadly with other countries/settings since this is typical age group. Weight by age if necessary if the age distribution of the available data is not flat.

11. In tables 2-3, show carriage results by smaller age strata since there are large differences in the age distribution by survey which confound the results. I suggest show vaccinated children for ages 3-4

years and 5-7 years separately, and for unvacc, perhaps 5, 6-7 and 8-10. This will enable comparisons across age groups within a time period, and across time periods for the same age group.

12. I did not understand the model methods in Suppl 1. This could be my limitations and might be fine, but it is not understandable to a non-biostatistician epidemiologist if that is a goal for either the Journal or the Authors.

13. The discussion needs assistance from a senior writer and a pneumococcal expert. From reviewing only the first two paragraphs the following points were observed:

- a. The most interesting observation is the prevalence of VT carriage (beyond ST3) in all groups studied, 3-6 years post PCV13 intro. This should be highlighted first.
- b. An '18% residual aggregated VT carriage' (line 330) is presented without mentioning the age distribution it represents, but the age group matters greatly, as the investigators have shown. This should be presented as <5 carriage or something other readers are familiar with, or can describe for several age groups (e.g., 18w-1y, 2y, 3-4y).
- c. The decline 'over the 3.5 year study period' (lines 325-326) needs to be contextualized in that it represents the time period 3-6 years post PCV13, since most decline will occur during the first 3 years.
- d. 'There was a modest increase in the NVT carriage.' (line 329) The magnitude is because of mathematics, not epidemiology, and so is not interesting. For example, because there is complete replacement in carriage, a 4% change (from 19% to 15%) in VT, which is a -21% relative change (4%/19%), there will be a 4% increase in NVT (from 58% to 62%), which is a +7% relative change (4%/58%).
- e. Same inappropriate comparison to VT carriage using PCV7 in HICs is made as in background (see 3f)
- f. Lines 338-342 – goes beyond limits of data, does not consider the influence of indirect effects of vaccination, speculative, not supported by analyses conducted in this paper.
- g. Lines 342-344: 'found a more marked decline..' – but the data are confounded by time since declines will be greater in the earlier years of PCV use than later, and %change will be greater in groups with higher %VT (younger ages, HIV+) and the groups do not have comparability in age distribution. So inferring comparisons in %change between groups is not advised without thorough discussion as to why.
- h. Lines 349-353: goes beyond limits of data, does not consider the influence of indirect effects of vaccination, speculative, not supported by analyses conducted in this paper. I also disagree with statement that effect of vaccination last only for first 6 months of life on the basis that vacc and unvacc children are similarly exposed pneumococcus yet the %carriage is lower in vaccinated children at any given age in this study. If there was no protection past 6 months of age, then all children >6m would have same %carriage regardless of vaccination status.

Other:

14. Throughout the paper, whenever referring to the adults, use 'HIV+ adults' to prevent misunderstanding since the previous Malawi study evaluated HIV- mothers and other studies often study HIV-negative populations or the general pop, not this special pop.

15. The carriage surveys are stated to be cross-sectional, yet the dates of NP collection are continuous (only the month of Sept 2015 were surveys not conducted). Please describe in more detail the methods to justify this. Otherwise, I suggest the language be changed to

16. Refusal rates were about 1%. This is shockingly low and does not seem plausible given the recruitment strategies. For the school-enrolled children, parents had to come to the school to be informed and agree – I suspect the refusal rates do not include parents who never came to the school. And for household recruitment, there was little detail on how eligible children were determined to be at home, who was present with authority to consent, etc. These rates are among a subset that met other 'screening' criteria first. Please include more information on what the 'screening' criteria were, including the numbers randomly selected to see what percent were screened, so that potential for bias can be more adequately assessed.

17. Was information on number of children <5 living in the household collected? This is a greater risk

factor for carriage than the other presented in Table 1.

18. Table 1

a. Describe where children were recruited from (household vs school)

b. describe the number of doses of PCV received.

c. Statistically compare at least some of the factors, such as the household characteristics, between relevant groups (3-7 year olds that were vaccinated vs 3-10 year olds that were not).

19. Describe HIV prevalence in this community in the background section. Children born to HIV+ mothers, even if not positive themselves, are thought to be at higher risk of disease. This may be relevant for effectiveness of PCV and VT carriage as well.

20. Rather than describe changes by survey number, translate this and use time since PCV introduction. Survey 1 is 3.5 years post-PCV intro and is important for the reader to always keep in mind when thinking about absolute carriage level and the amount of change that is still possible after carriage has already likely been reduced a lot from pre-PCV levels during the first 3 years of PCV use.

21. Give the units of age used in adjustment (months? Years?)

22. Suppl Fig 2:

a. separate out 6-8 wks olds

b. give denominator Ns in each bar (I suspect the 100% NVT in group age 3-<4 unvacc is n=1), or give 95%CI around %VT

c. explain notation on x-axis '(' and `]' in footnote

23. suppl 2a, Table 1: add Ns tested at each survey for each age group and split up into narrower age ranges as described elsewhere.

24. figure 2: the downward trend in %VT by survey in unvacc children 3-10 years is likely affected by differences in the age distribution over time. Show by narrower age ranges (3-4, 5-7, 8-10) to remove this potential age confounding. Do same for vacc children 3-7 years. Otherwise, the quick reader will likely compare changes over time between 3-7y vacc to 3-10y unvacc without considering the important differences in the age distribution between them, and likely make inappropriate conclusions.

25. figure 3:

a. the y-axis is VT distribution, not prevalence.

b. Split 3-7y and 3-10y into narrower age ranges (3-4, 5-7, 8-10)

c. Describe what the denominator of the percent is (I assume among VT isolates)

26. Fig 5:

a. there is likely some effect of 1-2 PCV doses on VT carriage, not zero as drawn.

b. The vaccinated arm is natural immunity + vaccine-induced, not natural imm only, otherwise the lines would be on top of each other at age 12+ months.

Reviewer #2:

Remarks to the Author:

Overview of the paper

The article presents time series surveillance data from a region of Malawi post-PCV introduction. The authors present VT and NVT prevalence data and report changes in prevalence over time, and carriage prevalence ratios calculated using log-binomial regression. The authors also use non-linear regression to estimate VT carriage decay rate and half life. They suggest that the program has not been as successful in Malawi as high-income settings, and has lead to reservoirs of infection in different sub-populations as well as low level immunity after the first year of life. They suggest that the 3+0 schedule approach should be changed and alternative vaccination approaches should be considered. The study is interesting, and presents carefully collected data analysed in a statistically robust manner.

I have a number of comments on the manuscript.

In general, at the moment the article assumes quite a high baseline level of pneumococcal epidemiological knowledge. A range of VT prevalence values are provided for pre PCV in sub-Saharan Africa, the authors then simply state that this is higher than high income settings but don't provide any values, as part of the point of the paper as I understand is to demonstrate that the program in Malawi has not worked well as these high income settings it would be helpful to provide the readership with some prevalences for high income settings. As this targeted for general readership it would be helpful to have some more explicit comparisons.

It is stated in the methods that if individuals in the study have one dose of vaccination individuals are considered to be vaccinated, however efficacy following the first dose is known to wane rapidly. It would be helpful if the authors could provide some rationale for why this was selected. Some sensitivity appears to have been conducted and they report that amongst those with a health passport 94.5% received 3 doses and only a small fraction had only 1 dose. However only 33% of participants provided a health passport. This leaves a high proportion of the children assigned as vaccinated through questionnaire only, I am unclear as to how reliable this is. Is it possible to conduct some additional sensitivity analyses on this?

The authors state that during the sample collection and serotyping they were able to differentially identify PCV13 serotypes within the data, but not NVTs. It's stated that there were not changes in the VT distribution observed as carriage declined. It would be helpful if the reader could see some graphs of the distribution of VT types over the time course. Equally it is possible that the distribution of VT prevalence did change after the introduction of vaccination in Nov 2011, but not 2.5+ years post-vaccination when sampling for this study was conducted.

On page 11 the authors report pneumococcal carriage. Reporting on small reductions in VT prevalence and providing NVT prevalence as well. In contrast to many regions in the world the NVT prevalence is exceptionally high, substantially higher than VT in all age groups. However this is not really remarked on by the authors. Some additional discussion should be given to this. Furthermore some examples of how this observation compares to NVT prevalence in high-income settings would be helpful to allow direct comparison of the situations.

Additionally despite high NVT prevalence VTs are still persisting, could this suggest some ongoing fitness advantage as they have yet to be outcompeted, or just due to the remaining high VT FOI?

One key factor that determines the success of vaccination programs is achieving a given level of coverage. Whilst the authors report that high schedule adherence and coverage were achieved I can't seem to see any data within the article to support this. Are any available? Or could it be that lower coverage has led to a less profound impact of vaccination, not just the schedule itself?

Lines 334-336. I'm not entirely sure it is fair to directly compare the impact of PCV vaccination between high-income and sub-Saharan African settings. Given they are slightly different transmission settings – equally the schedules have been different in high-income settings and the social contact patterns, healthcare provisions and general ecology for transmission are likely to differ markedly between the settings. Many factors between these settings differ. Some additional discussion on this should be provided.

372-379 – for the general readership it would be helpful if the authors could state which schedules

have been used in different African settings

Some discussion is provided on catch-up campaigns and their importance in helping to reduce overall transmission is widely acknowledged. Malawi did one when the campaign which was rolled out those <1, but the age groups included in other settings has been much larger, this should be described in further detail, particularly in comparison to high-income countries.

Lines 432-433 – it should also be acknowledged that not only will this lead to a potential underestimate of VT FOI but also ongoing NVT FOI as well.

As the authors acknowledge, given the high NVT prevalence already in the community and the 2.5+ year surveillance occurred after vaccination it is very possible serotype replacement had already occurred (given the timeline is about 3 years). However given the lack of pre-vaccination surveillance data this is not possible to validate.

Ultimately the biggest limitation to the paper and its findings is the lack of pre-vaccination data collected before Nov 2011 in this area. The authors wish to highlight that the program has not been as impactful in other settings at least in part to the 3+0 schedule used. However as we don't know what the original level of VT prevalence is it is somewhat difficult to confirm this using only their current data.

Point-by-point response to the referees' comments, Reviewer #1 Manuscript NCOMMS-19-19575.

Note: We have provided line numbers to show where text in the manuscript has been adjusted in response to reviewer's comments. The line numbers refer to the Clean/Untracked versions of submitted Word files.

This study shows the vaccine-type (VT) carriage remaining after 3-6 years of PCV13 use in an urban area in Malawi. It describes carriage in vaccinated and unvaccinated children and unvaccinated HIV+ adults in a setting with high PCV coverage using a 3+0 schedule under conditions of high pneumococcal carriage. This study adds to the literature on the epidemiology of pneumococcal carriage after long-term use of PCV in low-income settings.

- *However, the paper needs some attention from more senior, experienced writers and pneumococcal experts since the evidence shown is insufficient to support many of the claims made by the authors and inferences on mechanisms of action of immunity, specifically vaccine-induced vs immunity induced by natural exposure, and on duration of vaccine-induced immunity go beyond the limits of the data.* The below items include many issues that need to be addressed in this manuscript, but likely not all as I stopped at line 353 when it was clear that more senior guidance in general is required.

Response: Overall, the comments made by Reviewer #1 help improve the clarity of the manuscript. However, Reviewer #1 has unfairly characterised the authorship of this manuscript. Heyderman, French, Everett and Bar-Zeev together have a considerable international standing in the pneumococcal field as evidenced by numerous impactful publications over the last 20 years, and membership of relevant grant, specialist society and public health advisory committees. Gupta and Diggle are internationally recognised leaders in the field of disease transmission modelling. The question of expertise or speculation has not been raised by Reviewer #2. As demonstrated by our responses to Reviewer #1, all the comments made can be easily addressed, and the inferences on mechanisms are based on the results of our analysis, supported by a considerable body of evidence generated by others. In the revision we have made clear where we have used speculation.

- *The below items include many issues that need to be addressed in this manuscript, but likely not all as I stopped at line 353 when it was clear that more senior guidance in general is required.*

Response: We have comprehensively addressed the very helpful comments made by Reviewer #1 below. Line 353 is midway through the discussion. We have made sure that the remainder of the discussion addresses the Reviewer's concerns made in the other section of the manuscript.

Major items to address

Results:

1. *PCV13 is not effective in preventing ST3 carriage, so also estimate impact counting ST3 as a non-vaccine-preventable ST. This will restrict analyses to the STs that are vaccine-preventable. This eliminates any bias when comparing settings on the contributions of ST3*

Response: We acknowledge that several post-PCV13 introduction studies have demonstrated the poor vaccine efficacy against ST3 carriage and disease. However, to maintain comparability with other carriage studies, we have retained the grouping together of all vaccine serotypes in our analysis. To address the Reviewer’s concern, we have assessed this potential bias by removing ST3 from the analysis and have reported the results in the supplementary material (Supplementary Table 9). This re-analysis did not result in substantial changes in the trends seen in VT prevalence except for the adjusted prevalence ratios (aPR) for VT carriage prevalence among PCV-vaccinated children 3-5 years of age which followed the same trend but was no longer statistically significant: aPR 0.919 [95% CI 0.845, 0.999] $p = 0.047$ for analysis including ST3 vs. aPR 0.942 [95% CI 0.856, 1.036] $p = 0.218$ for analysis not including ST3. We have included a line in the text in reference to this additional analysis, citing manuscripts reporting the poor efficacy of PCV13 against serotype 3. (Lines: 341-348)

2. Abstract conclusions

Line 23: *The wording “compared to HICs, the 3+0 schedule in Malawi has led to a suboptimal reduction in pneumococcal carriage prevalence” (1) goes beyond the scope of the data (no HICs were studied) and (2) implies any differences are due to the schedule, not the setting (higher overall spn carriage, higher prevalence of risk factors, higher HIV prevalence, etc.).*

Response: We agree with the reviewer that this sentence is potentially misleading. We have modified this sentence to “Compared to high-income settings, there is a high residual prevalence of vaccine serotype *S. pneumoniae* carriage up to 7 years after introduction of PCV in Malawi.” (Lines: 23-24)

Line 24: *The wording “This is likely due to recolonisation of vaccinated children with waning vaccine-induced immunity.” is speculative and not supported by the data. Conclusions should be based only on the evidence in the data; any speculations must be worded as such.*

Response: We agree with the reviewer that this sentence is potentially misleading. We have modified this sentence to “We propose that a key parameter in high residual carriage is a high force of infection that drives recolonisation among children with waning vaccine-induced immunity, further allowing insufficient indirect protection of PCV-unvaccinated older children and HIV-infected adults.” (Lines: 24-27)

Background

3. Background needs attention from a more experienced investigator. It reads as if it was not thoughtfully composed. Examples of issues include:

a. *the pneumococcal deaths estimate cited is for the pre-PCV era and two decades ago (2000). Deaths have come down considerably since then. Use more recent (2015) estimates (B. Wahl, Lancet, 2018).*

Response: Our initial draft of the manuscript was written prior to the Wahl publication, we should have updated it. This section has now been updated with the Wahl publication. (Lines: 35-37)

b. *results of the previous post-PCV carriage study in Malawi are not described, only their pre-PCV results (ref #11). This is important because need to show what has already been done and to justify why another carriage study is needed. (I think it*

is - because the other study was early post-PCV (yr 2 post), had small N, and adults were HIV- mothers of infants rather than HIV+ adults.)

Response: We have provided further description of this study in the Background (Lines: 97-100) as well as a table in the supplement (Supplementary Table 5) that allows the reader to see the similarities in carriage dynamics between both the Karonga and Blantyre sites.

- c. *“differing markedly from high-income settings” – only cited African LIC data but need to support this statement by also citing HIC data.*

Response: The additional references have been added (see also response to comment h). (Lines: 64-66)

- d. *Since ‘markedly’ is subjective, need to also state the range of VT carriage in HICs to compare to the estimates provide for LICs.*

Response: These have now been added (see also response to comment h) (Lines: 64-66)

- e. *Add ref 19 to ref list ‘15-18’ on line 55*

Response: This has been done, though reference listing has changed slightly. (Line: 70)

- f. *The residual carriage estimates cited (refs 22-24, line 57) include only one PCV10/13 using country; the other studies are impact with PCV7. Impact between HICs and LICs should be compared using the same serotypes since*

- a. *(1) PCV10/13 ‘VT’ carriage is higher because it has more STs, and*
- b. *(2) PCV7 was effective in preventing all STs it covered whereas PCV13 does not protect against ST3 carriage (and probably slow-acting if any efficacy against IPD).*

Response: We agree with the reviewer that these data should ideally compare impact of PCV based on PCV of the same valency. For this reason, we have added text to acknowledge the difficulty in interpreting data comparing countries and regions with a history different PCV implementation strategies. We provide i) background on differences in PCV introduction in the Introduction (Lines: 47-56; 70-75) and ii) more detailed overview of the different vaccines and schedules that have been implemented in Supplementary Table 1.

- g. *Line 57: the claim of rapid onset of NVT replacement in carriage in the Africa region implies it does not occur everywhere – this is false. NVT replacement is a known phenomenon of spn carriage everywhere.*

Response: To avoid any misunderstanding, we have adjusted the wording to clarify that NVT replacement is not unique to the Africa region. (Lines: 80-81)

- h. *Line 58-60: this sentence implies carriage replacement with NVT types means disease replacement. This is only true if they are virulent like the VTs, but they are not equally so. Read the literature on IPD replacement with NVTs in HICs and LICs and revise.*

Response: We are well aware of the literature on the impact of VT replacement by NVT. The complexity of this impact is not explained simply by the invasive potential of the replacing serotypes and also appears to be different between populations even in HICs. Given that the main conclusions of this manuscript are focused on VT, we did not review this complexity in the original version. However, to address the Reviewer's concerns, we provided a more complete picture. (Lines: 81-84)

- i. *Lines 60-62: this sentence about 3+0 schedule is a non-sequitur. There is no mention of other schedule choices and what would be preferred. Or what this schedule has to do with anything discussed so far.*

Response: We clarified this section, introducing text that compares PCV implementation in HICs (e.g. usually including booster doses) to the approach in LMICs in SSA (predominantly without booster doses). (Lines: 47-58)

4. The PCV coverage in this community in inadequately described and the description of the 4a. validation method for determining vaccination status of older kids (which only 25% had immunization card) is insufficient. Only 60 children were evaluated and the number that were not vaccinated, which is the validation in question, is not described. Nor were any details on the sensitivity and specificity of the findings presented. The level and potential for misclassification of vaccination status is important to understand since this defines the primary analysis comparison groups.

Response: We recruited children whose parents reported a history of being vaccinated. In the context of a high vaccine uptake (as reported by the Malawi national immunisation programme), parental recall is likely to be highly predictive. In our previous population-based vaccine studies, [References 1 & 2, below] we have previously found excellent agreement between hand-held vaccine records and parental reporting [unpublished]. Confirmation using EPI records at the health centre level is not possible in this setting because of the methods of documentation. We accept that misclassification could have occurred but based on our previous data and the additional validation included in the present study, we maintain that this misclassification is likely to be small and would not change the interpretation of the results. We have presented the level of agreement between the history of vaccination and the hand-held record but have not calculated specificity and sensitivity, as this is not a standard approach. To address the Reviewer's concerns, we have further addressed vaccine coverage in the Limitations section (Lines: 488-499)

4b. The vaccination coverage in infants over the past years in this community is also important since it describes the conditions in which this carriage data represent. Other evidence should be given to describe the annual coverage estimates rather than parent recall of something that happened many years ago. I do not think it is reasonable to assume that parents of a 7 year old remember what legs/arms were jabbed, how many shots, etc., their child received 7 years ago. They may be remembering even the vaccination of a younger sibling, if at all.

Response The vaccine coverage that we report is similar to both recent UNICEF data and other peer-reviewed articles (see response 4a). We also report high BCG vaccination levels (as evidenced by confirmed BCG scars) and report work done by Jahn et al. in using scar data

as inference of population vaccination coverage independently from vaccination records. This has been addressed in the Limitations section. (Lines: 488-499)

5. Since the analysis compares changes in VT carriage over time (Tables 2-3) and by age (Figure 4), describing any differences in the Table 1 characteristics by these factors would show whether any are potential confounders.

Response: As suggested, we have assessed whether these individual covariates have changed over the study period. Although there was change in some covariates (Crowding index, Smoker in household, Latrine type, Electricity at household, and Possessions index), the magnitude was limited. Furthermore, when adjusted for in the model as single covariates, they made no meaningful difference to the relationship between VT carriage and time. While multiple models were investigated, the simplest models consistently confirmed a slow decline over time. Increasing the complexity of these models reported a very similar outcome (a slow decline over time) but consistently lost their statistical significance with increased degrees of freedom and, consequently, widening confidence intervals. We have reported this in the manuscript (Lines: 333-339)

6. Describe the units for the prevalence ratio – is this comparing month to month?? It would more informative if shown as annual change (change over 12 months).

Response: As recommended by the Reviewer, this has been recalculated to show as annual change. The description of these methods has been adjusted accordingly (Lines: 207-210)

7. Line 259: authors state the reduction in VT carriage among vaccinated children was not exponential and cite Suppl 1. Yet I could not find where this decay function was shown to be ruled out. I suspect they did not have the statistical power to rule out this possibility, especially given they are likely at the tail end of the impact since the study started after 3.5 years of PCV use, after when the majority of the decline would be expected.

Response: When we modelled the entire range of data (i.e. not censoring at 3.6 years) with an exponential decay curve, the estimated decay parameter for vaccinated was not significantly different from zero, i.e. an almost flat curve was the resulting fit. Moreover, Figure 2 of supplement 1 (now supplementary figure 1) shows what could be considered a visual, discrete version of the model fitted to the data. There is an observed increase in pneumococcal carriage until the 3-4 year age range and only then does VT carriage prevalence start to decrease. Our conjecture is that the overall trend in pneumococcal carriage for vaccinated children starting from birth is more complex than an exponential decay curve. However, the data are too sparse to robustly fit such a complex relationship.

In addition, when referencing Supplement 1 (now Supplementary Figure 1) in the manuscript, we refer the reader specifically to the Supplementary Figure 1, which has additional text to address the observation on non-exponential decay.

8a. The study is completely confounded by age and year post-PCV intro as there is no meaningful overlap in any survey period of vaccinated and unvaccinated children of the same age. This means effects of age and time cannot disentangled and so no assumptions about differences in VT carriage can be attributed to either age or time.

Response: We do not agree with the Reviewer and this wide-ranging conclusion was not made by Reviewer #2. In relation to the entire analysis, because age is a (likely) confounder

within each target group, we do control for age in the primary analysis (adjusted prevalence ratio, aPR). We agree that there is limited overlap however the intent is not to compare carriage prevalence between vaccinated and unvaccinated children to define, for example, a vaccine effect size which we agree would be problematic. The intention is to define the trend in carriage over time within each study group in the context of a vaccine programme with high coverage: younger vaccinated children (benefiting from direct vaccine-induced protection), older unvaccinated children (not benefiting from direct vaccine-induced protection), and HIV-infected adults on ART (not benefiting from direct vaccine-induced protection).

In relation to the Statistical model (Non-linear regression analysis for VT carriage decay rate and half-life), the data is not analysed by survey period and overall there is an overlap of vaccinated and unvaccinated children of the same age. If this were not the case, we would have not been able to fit the model using all the data and would have been constrained to fit two separate exponential decay models to vaccinated and unvaccinated children. Moreover, we clearly report the limitation of minimal overlap in age in both the Supplementary material and the Discussion. (Lines: 483-486)

8b. One suggestion, if there is sufficient sample size, is to show VT carriage in Figure 4 by survey (i.e., multiple red lines and multiple blue lines) and restrict to ages the vaccinated and unvaccinated populations are assessed in most time periods. For vaccinated (blue line) this would be ages 4-5 years and ages 6-7 years for unvac (red). This would eliminate the confounding the other ages contribute to vacc and unvac groups since they differ greatly in their age distribution. Then can see if there is a smooth transition between blue 5 to red 6 years or if there appears to be a jump relative to the changes from blue 4-5 and red 6-7 in any given time period.

Response: We believe this would not be an informative approach for the non-linear regression analysis, as the data are already quite sparse and sample size would not be sufficient to fit separate curves for vaccinated and unvaccinated children for each survey period. Moreover, constraining the analysis to those age ranges would force us to discard a considerable amount of useful information.

8c. Regardless, because of this age-time confounding problem, the results and conclusions should not focus in this analysis, but rather on the more interesting long-term VT carriage prevalence observed. The decay with age is well known and has been documented pre-PCV so that analysis adds nothing there. And the rate or shape of the decline has not been made relevant to understand from a policy point of view why we should care about it. Perhaps it could be used to speculate on how long it will take for elimination of VT carriage in this community, but I think that goes beyond the limits of this data.

Response: In the revised manuscript we have ensured that there is a clear emphasis on the high residual VT carriage since the 2011 PCV introduction. However, the rate of decay in VT carriage is also of considerable public health interest. Our methodological approach allows us to better understand the contribution of vaccine-induced immunity in reducing the risk of carriage (compared to those older children who were age-ineligible for PCV) as natural immunity develops and plays a larger role in carriage control. As the Reviewer rightly suggests, using the slope to speculate how long it will take for elimination of VT carriage 'goes beyond the limits of this data.' However, the slope does show that there is a temporal

change in VT carriage. As such, we have included further discussion of whether alternative vaccine schedules with a booster vaccine dose for example, can be used to increase the rate in VT carriage decay.

9. *Methods for calculating 'relative change' in Table 2-3 is not described. This should also be presented with CIs.*

Response: The description for calculating 'relative change' has been added (Lines: 216-218). In reviewing similar manuscripts, the relative change is usually not provided and where provided, a confidence interval is not included. The primary analysis of interest is the reported prevalence ratios and these are provided with the necessary confidence interval. We have therefore not included confidence intervals for the reported relative change.

10. *For external comparability with the previous NP carriage study done in Malawi, present carriage results for 1-4year olds (as opposed to only stratified 18k-1 year and 2 years and 3-7/10 years) and show for <5 year olds (18wk-4y) for external comparability more broadly with other countries/settings since this is typical age group. Weight by age if necessary if the age distribution of the available data is not flat.*

Response: We have provided further description of this study in the Background (Lines: 97-100) as well as a table in the supplement (Supplementary Table 5). Weight by age was not necessary given the age distribution.

11. *In tables 2-3, show carriage results by smaller age strata since there are large differences in the age distribution by survey which confound the results. I suggest show vaccinated children for ages 3-4 years and 5-7 years separately, and for unvacc, perhaps 5, 6-7 and 8-10. This will enable comparisons across age groups within a time period, and across time periods for the same age group.*

Response: We completed a preliminary analysis using the stratification recommended by the reviewer. However, this stratification does not fit well within our sampling frame and the evolving vaccine coverage over time. This approach resulted in small group sizes that do not allow robust conclusions to be drawn. We have therefore not include the output in the main manuscript or supplementary information. We have, however, included the output as a **supporting document** with the re-submission for the Reviewer and the Editor.

However, as an alternative approach, we have restricted the ages of children included in the prevalence ratio (aPR) analyses. We have limited the ages to 3-5 year old PCV-vaccinated children and 6-8 year old PCV-unvaccinated children. This, along with the already-incorporated controlling of age in the model as a likely confounder, will address the reviewer's concerns that there is a bias due to age. We have described this in the Methods section (Lines: 211-216). To provide the reader with the carriage data used in Figure 2, we have included the table submitted in our original submission (without the prevalence ratios) in Supplementary Tables 3 and 4.

12. *I did not understand the model methods in Suppl 1. This could be my limitations and might be fine, but it is not understandable to an non-biostatistician epidemiologist if that is a goal for either the Journal or the Authors.*

Response: We have adjusted text in both the Methods (Lines: 223-230), the Results and the Supplementary material pages 10-12 (formerly Suppl 1) in order to help the Reviewer and the readership better understand the modelling methods.

13. *The discussion needs assistance from a senior writer and a pneumococcal expert.*

Response: We remain puzzled by this unsubstantiated comment. All the comments made by the Reviewer are easily addressed.

From reviewing only the first two paragraphs the following points were observed:

a. *The most interesting observation is the prevalence of VT carriage (beyond ST3) in all groups studied, 3-6 years post PCV13 intro. This should be highlighted first.*

Response: We agree that the residual prevalence of VT carriage 3-6 years post PCV13 intro is a key observation. This finding has been more strongly emphasised, underlining the high residual carriage starting 3.6 years post EPI introduction and through until study end at -7.1 years post EPI introduction (Examples include: Abstract lines 14-15; Discussion lines; footnote to Figure 2)

b. An '18% residual aggregated VT carriage' (line 330) is presented without mentioning the age distribution it represents, but the age group matters greatly, as the investigators have shown. This should be presented as <5 carriage or something other readers are familiar with, or can describe for several age groups (e.g., 18w-1y, 2y, 3-4y).

Response. We have adjusted this to report the VT carriage prevalence among the same age stratum (1-4 years old) to which it is compared. (Lines: 387-389)

c. *The decline 'over the 3.5 year study period' (lines 325-326) needs to be contextualized in that it represents the time period 3-6 years post PCV13, since most decline will occur during the first 3 years.*

Response: Throughout the manuscript we have reiterated the fact that this study represents 3.6 to 7.1 years after Malawi's 12 November 2011 introduction of PCV. In the revised discussion we have made sure that this is clear (including lines 380-381 as a modification of the former lines 325-326). We have also clarified that there would have been a considerable decline in VT carriage before commencement of our surveys.

d. 'There was a modest increase in the NVT carriage.' (line 329) The magnitude is because of mathematics, not epidemiology, and so is not interesting. For example, because there is complete replacement in carriage, a 4% change (from 19% to 15%) in VT, which is a -21% relative change (4%/19%), there will be a 4% increase in NVT (from 58% to 62%), which is a +7% relative change (4%/58%).

Response: We disagree with Reviewer #1 that the change in NVT carriage is not interesting. The phrase that "there was only a modest increase in NVT carriage" has been used to emphasise that there has not been wholesale NVT replacement. Indeed, overall total pneumococcal carriage rates are falling. This has been further clarified in the text.

e. *Same inappropriate comparison to VT carriage using PCV7 in HICs is made as in background (see 3f)*

Response: This has been clarified in the response to 3f.

f. *Lines 338-342 – goes beyond limits of data, does not consider the influence of indirect effects of vaccination, speculative, not supported by analyses conducted in this paper*

Response: This section provides context for our findings. Where we mentioned “not simply vaccine dependent” we were suggesting that indirect effects alone do not explain our data. In the revised manuscript, we have provided support from the literature for the assertion that there is a “complex relationship in the first few years of life between VT carriage and the impact of waning vaccine-induced mucosal immunity and acquisition of natural immunity.” We have clarified to role of indirect protection in this complex process.

g. *Lines 342-344: ‘found a more marked decline..’ – but the data are confounded by time since declines will be greater in the earlier years of PCV use than later, and %change will be greater in groups with higher %VT (younger ages, HIV+) and the groups do not have comparability in age distribution. So, inferring comparisons in %change between groups is not advised without thorough discussion as to why.*

Response: We agree with the Reviewer and because these are not central finding, we have deleted this sentence from the manuscript.

h. *Lines 349-353: goes beyond limits of data, does not consider the influence of indirect effects of vaccination, speculative, not supported by analyses conducted in this paper.*

Response: The readership of this manuscript will look to the authors for interpretation of our key findings, but we agree that it should be evidence-based. In the revised manuscript, we have provided additional evidence from the literature to support the proposed mechanisms. (Lines; 405-410)

I also disagree with statement that effect of vaccination last only for first 6 months of life on the basis that vacc and unvacc children are similarly exposed pneumococcus yet the %carriage is lower in vaccinated children at any given age in this study. If there was no protection past 6 months of age, then all children >6m would have same %carriage regardless of vaccination status.

Response: We have not said that “vaccination lasts only for the first 6 months of life” and that there is “no protection past 6 months”. We have therefore, further clarified this sentence and provided further evidence from the literature to substantiate our statement that “much of that direct vaccine-induced protection against carriage occurs quite early, perhaps within the first 6 months of life”. (Lines; 405-410)

Other:

14. *Throughout the paper, whenever referring to the adults, use ‘HIV+ adults’ to prevent misunderstanding since the previous Malawi study evaluated HIV- mothers and other studies often study HIV-negative populations or the general pop, not this special pop.*

Response: We have adjusted text to ensure reference to HIV-infected adults when referring to the adult study population, or where otherwise appropriate.

15. *The carriage surveys are stated to be cross-sectional, yet the dates of NP collection are continuous (only the month of Sept 2015 were surveys not conducted). Please describe in more detail the methods to justify this. Otherwise, I suggest the language be changed to*

Response: We have adjusted the language, referring to rolling cross-sectional surveys. (Lines: Title, 9, 114, 284)

In addition, in the section 'Site Selection and Recruitment', we have provided additional context on activities that occurred before and/ or during each survey that established these as individual (rolling) surveys. (Lines: 141-142, 148-149)

16. Refusal rates were about 1%. This is shockingly low and does not seem plausible given the recruitment strategies. For the school-enrolled children, parents had to come to the school to be informed and agree – I suspect the refusal rates do not include parents who never came to the school.

Response: This has been clarified in the manuscript (Lines: 258-263) and added as a footnote to the recruitment flow-chart (Figure-1).

And for household recruitment, there was little detail on how eligible children were determined to be at home, who was present with authority to consent, etc. These rates are among a subset that met other 'screening' criteria first. Please include more information on what the 'screening' criteria were, including the numbers randomly selected to see what percent were screened, so that potential for bias can be more adequately assessed.

Response: To clarify screening criteria, how eligible children were determined to be at home and who had authority to consent, we have explained this further in the Methods section of the manuscript (Lines 145-153) and added as a note to the recruitment flow-chart (figure-1). We provided details on houses and individuals screened in the results section. (Lines: 258-264)

17. Was information on number of children <5 living in the household collected? This is a greater risk factor for carriage than the other presented in Table 1.

Response: This has been added to Table 1.

18. Table 1

a. Describe where children were recruited from (household vs school)

Response: In addition to the sentence "Among the children in the final analysis, 3605 were recruited from households and 1427 from schools" (Lines: 203-204 in original submission), further details on where children were recruited has been added to Table 1 as a footnote.

b. describe the number of doses of PCV received.

Response: This has been added to Table 1.

c. Statistically compare at least some of the factors, such as the household characteristics, between relevant groups (3-7 year olds that were vaccinated vs 3-10 year olds that were not).

Response: We have provided this analysis, reporting those that are statistically significant in the footnote of Table 1. We have described how the comparison was made in the Methods section (Lines: 203-205). We have also worked to further clarify for the reader that we are not comparing trends in carriage prevalence between vaccinated and unvaccinated children to define a vaccine effect size, which we agree would be problematic. In describing the sampling frame (Site Selection and Recruitment) for each group, we acknowledge that

target groups were not recruited using the same method. The intention of the study is to define the trend in carriage over time within each study group in the context of a well-run vaccine programme.

19. Describe HIV prevalence in this community in the background section. Children born to HIV+ mothers, even if not positive themselves, are thought to be at higher risk of disease. This may be relevant for effectiveness of PCV and VT carriage as well.

Response: We have shown previously, as have others, that HIV exposure is not a risk factor for carriage (Heinsbroek Am J Epi 2015). However, we have included more details on HIV prevalence in the Background (Lines: 90-95).

20. Rather than describe changes by survey number, translate this and use time since PCV introduction. Survey 1 is 3.5 years post-PCV intro and is important for the reader to always keep in mind when thinking about absolute carriage level and the amount of change that is still possible after carriage has already likely been reduced a lot from pre-PCV levels during the first 3 years of PCV use.

Response: While maintaining the use of survey number as primary unit of time sequence, we have introduced text to emphasise how this translates into 'time since Malawi's November 2011 introduction of PCV (3.6-7.1 years since PCV introduction).

21. Give the units of age used in adjustment (months? Years?)

Response (Done): We have reported this as years (Line: 318)

22. Suppl Fig 2:

a. separate out 6-8 wks olds

Response: We have added a footnote to the figure clarifying that the (0, 1] bar among unvaccinated children includes only children 4-8 weeks of age.

b. give denominator Ns in each bar (I suspect the 100% NVT in group age 3-<4 unvacc is n=1), or give 95%CI around %VT

Response: We have reported the denominators for the figures in a table below the figure.

c. explain notation on x-axis (' and 'j' in footnote

Response: The notation has been included in the footnote

23. suppl 2a, Table 1: add Ns tested at each survey for each age group and split up into narrower age ranges as described elsewhere.

Response: To minimise repetition, within the supporting text for supplementary Tables 2a and 2b (now supplementary tables 7 and 8), we have referred the reader to supplementary tables 3 and 4 which report 'Ns tested at each survey.' The use of different age ranges has been discussed above.

24. figure 2: the downward trend in %VT by survey in unvacc children 3-10 years is likely affected by differences in the age distribution over time. Show by narrower age ranges (3-4, 5-7, 8-10) to remove this potential age confounding. Do same for vacc children 3-7 years. Otherwise, the quick reader will likely compare changes over time between 3-7y vacc to 3-

10y unvacc without considering the important differences in the age distribution between them, and likely make inappropriate conclusions.

Response: As discussed above, the stratification recommended was not informative and we have therefore decided to not include the output.

However, as reported for Question 11, we have restricted the ages of children included in the prevalence ratio analyses to 3-5 year old PCV-vaccinated children and 6-8 year old PCV-unvaccinated children. This, along with the already-existing controlling of age in the model as a likely confounder, will address this potential bias.

25. figure 3:

a. the y-axis is VT distribution, not prevalence.

Response: This has been corrected

b. Split 3-7y and 3-10y into narrower age ranges (3-4, 5-7, 8-10)

Response: This has been discussed above

c. Describe what the denominator of the percent is (I assume among VT isolates)

Response: A description has been included, clarifying that it is 'among VT isolates'

26. Fig 5:

a. there is likely some effect of 1-2 PCV doses on VT carriage, not zero as drawn.

Response: We have adjusted the figure to show that, as Reviewer #1 rightly suggests, PCV doses 1-& 2 do have some effect in controlling carriage.

b. The vaccinated arm is natural immunity + vaccine-induced, not natural imm only, otherwise the lines would be on top of each other at age 12+ months.

Response: We have adjusted the figure to include Natural + Herd + Vaccine-induced immunities for the vaccinated arm. We have included Natural + Herd immunities for the unvaccinated arm.

References

- 1 Bar-Zeev N, Kapanda L, Tate JE, Jere KC, *et al.* Effectiveness of a monovalent rotavirus vaccine in infants in Malawi after programmatic roll-out: an observational and case-control study. *Lancet Infect Dis.* 2015;15:422-8.
- 2 Mvula H, Heinsbroek E, Chihana M, *et al.* Predictors of Uptake and Timeliness of Newly Introduced Pneumococcal and Rotavirus Vaccines, and of Measles Vaccine in Rural Malawi: A Population Cohort Study. *PLoS One* 2016;11: e0154997.
- 3 Sigaúque B, Moiane B, Massora S, *et al.* Early Declines in Vaccine Type Pneumococcal Carriage in Children Less Than 5 Years Old After Introduction of 10-valent Pneumococcal Conjugate Vaccine in Mozambique. *Pediatr Infect Dis J* 2018;37:1054–1060
- 4 Hammitt LL, Etyang AO, Morpeth SC, *et al.* Effect of ten-valent pneumococcal conjugate vaccine on invasive pneumococcal disease and nasopharyngeal carriage in Kenya: a longitudinal surveillance study. *Lancet* 2019; 393: 2146–54. [http://dx.doi.org/10.1016/S0140-6736\(18\)33005-8](http://dx.doi.org/10.1016/S0140-6736(18)33005-8)

Point-by-point response to the referees' comments, Reviewer #2 Manuscript NCOMMS-19-19575.

Note: Below is a point by point response to Reviewer #2. Changes to the manuscript have also been made in response to the comments from another Reviewer. These are detailed in an appendix (attached).

Note: We have provided line numbers to show where text in the manuscript has been adjusted in response to reviewer's comments. The line numbers refer to the Clean/Untracked versions of submitted Word files.

Overview of the paper

The article presents time series surveillance data from a region of Malawi post-PCV introduction. The authors present VT and NVT prevalence data and report changes in prevalence over time, and carriage prevalence ratios calculated using log-binomial regression. The authors also use non-linear regression to estimate VT carriage decay rate and half life. They suggest that the program has not been as successful in Malawi as high-income settings, and has led to reservoirs of infection in different sub-populations as well as low level immunity after the first year of life. They suggest that the 3+0 schedule approach should be changed and alternative vaccination approaches should be considered. The study is interesting, and presents carefully collected data analysed in a statistically robust manner.

I have a number of comments on the manuscript. In general, at the moment the article assumes quite a high baseline level of pneumococcal epidemiological knowledge.

- *A range of VT prevalence values are provided for pre PCV in sub-Saharan Africa, the authors then simply state that this is higher than high income settings but don't provide any values, as part of the point of the paper as I understand is to demonstrate that the program in Malawi has not worked well as these high income settings it would be helpful to provide the readership with some prevalences for high income settings. As this targeted for general readership it would be helpful to have some more explicit comparisons.*

Response: We have added details of carriage prevalence in high income settings pre-dating PCV introduction. (Lines 64-66). We have also added an additional table (Supplementary Table 2) providing an overview of the literature on residual carriage after PCV introduction in high- and low-income country settings.

- *It is stated in the methods that if individuals in the study have one dose of vaccination individuals are considered to be vaccinated, however efficacy following the first dose is known to wane rapidly. It would be helpful if the authors could provide some rationale for why this was selected. Some sensitivity appears to have been conducted and they report that amongst those with a health passport 94.5% received 3 doses and only a small fraction had only 1 dose. However only 33% of participants provided a health passport. This leaves a high proportion of the children assigned as vaccinated through questionnaire only, I am unclear as to how reliable this is. Is it possible to conduct some additional sensitivity analyses on this?*

Response: One dose was used to achieve a high sensitivity for vaccine exposure. The sensitivity analyses performed do address Reviewer #2's concerns about waning immunity.

Regarding the reliability of our vaccine coverage estimates and risk of misclassification, we recruited children whose parents reported a history of being vaccinated. The vaccine coverage that we report is similar to both recent UNICEF data and other peer-reviewed articles (Referenced in manuscript, Lines 89-90).

We also report high BCG vaccination levels (as evidenced by confirmed BCG scars) and report work done by Jahn et al. in using scar data as inference of population vaccination coverage independently from vaccination records. To address the Reviewer's concerns, we have further addressed vaccine coverage (with references) in the Limitations section (Lines: 488-499)

In the context of a high vaccine uptake, parental recall is likely to be highly predictive. In our previous population-based vaccine studies, (References 1 & 2, below) we have previously found excellent agreement between hand-held vaccine records and parental reporting [unpublished]. Confirmation using EPI records at the health centre level is not possible in this setting because of the methods of documentation. We accept that misclassification could have occurred but based on our previous data and the additional validation included in the present study, we maintain that this misclassification is likely to be small and would not change the interpretation of the results.

- *The authors state that during the sample collection and serotyping they were able to differentially identify PCV13 serotypes within the data, but not NVTs. It's stated that there were not changes in the VT distribution observed as carriage declined. It would be helpful if the reader could see some graphs of the distribution of VT types over the time course.*

Response: Though not presented as a graph, this information had been provided in the original supplementary material as a table in Supplementary tables 2a and 2b. We have included the same data as supplementary tables 7 and 8 of the resubmission.

- *Equally it is possible that the distribution of VT prevalence did change after the introduction of vaccination in Nov 2011, but not 2.5+ years post-vaccination when sampling for this study was conducted.*

Response: We have discussed this point in the Limitations section, regarding re-adjusted carriage dynamics (VT & NVT prevalence as well as serotype-specific trends) that might have occurred between PCV introduction and our first survey. (Lines: 485-487)

- *On page 11 the authors report pneumococcal carriage. Reporting on small reductions in VT prevalence and providing NVT prevalence as well. In contrast to many regions in the world the NVT prevalence is exceptionally high, substantially higher than VT in all age groups. However, this is not really remarked on by the authors. Some additional discussion should be given to this. Furthermore some examples of how this observation compares to NVT prevalence in high-income settings would be helpful to allow direct comparison of the situations.*

Response: In low income country settings (LICs), similar prevalences of pneumococcal carriage, including an NVT prevalence higher than VT prevalence, is not exceptional (References 3 & 4, below). It is true that NVT are less prominent in disease.

- *Additionally despite high NVT prevalence VTs are still persisting, could this suggest some ongoing fitness advantage as they have yet to be outcompeted, or just due to the remaining high VT FOI?*

Response: we agree with the Reviewer that it is likely that both VT fitness and force of infection play a role. But at this stage we can only speculate without evidence and therefore have not included this in our discussion. Outside the remit of this manuscript, our whole genome sequencing analysis shows that there are NVTs emerging that have acquired virulence characteristics and antimicrobial resistance that may give them a competitive advantage. This will be submitted for publication in due course.

- *One key factor that determines the success of vaccination programs is achieving a given level of coverage. Whilst the authors report that high schedule adherence and coverage were achieved I can't seem to see any data within the article to support this. Are any available? Or could it be that lower coverage has led to a less profound impact of vaccination, not just the schedule itself?*

Response: For this study population, we do report PCV13 coverage as 98.7% among those screened and age-eligible for PCV vaccination. (Lines 273-274).

We have also included reference to previous work on vaccine coverage in Malawi, referencing two previous studies that report similarly high PCV13 uptake rates of 90%–95%. In addition, we reference the 92% PCV13 coverage recently reported by WHO/UNICEF (Lines: 89-90)

- *Lines 334-336. I'm not entirely sure it is fair to directly compare the impact of PCV vaccination between high-income and sub-Saharan African settings. Given they are slightly different transmission settings – equally the schedules have been different in high-income settings and the social contact patterns, healthcare provisions and general ecology for transmission are likely to differ markedly between the settings. Many factors between these settings differ. Some additional discussion on this should be provided.*

Response: The aim of vaccine programmes in sSA is to achieve the same vaccine impact in LMICs as in HICs, which would require adequate herd immunity. We acknowledge the difficulty in interpreting data comparing countries, even within the same region, with a history of different PCV implementation strategies. In the Background, to underline this challenge, we have added details on differences in strategies between countries (Lines: 47-56; 70-75). In the supplement we have also added a more detailed overview of country-specific strategies (incl. vaccines valence, schedules, changes over time; Supplementary Table 1).

- *372-379 – for the general readership it would be helpful if the authors could state which schedules have been used in different African settings*

Response: We have added a table to the supplement which provides an overview of country-specific strategies (incl. vaccines valence, schedules, changes over time; Supplementary Table 1).

- *Some discussion is provided on catch-up campaigns and their importance in helping to reduce overall transmission is widely acknowledged. Malawi did one when the campaign which was rolled out those <1, but the age groups included in other settings has been much larger, this should be described in further detail, particularly in comparison to high-income countries.*

Response: While avoiding substantially lengthening the Discussion, we have added some additional detail. (Lines: 442-445)

- *Lines 432-433 – it should also be acknowledged that not only will this lead to a potential underestimate of VT FOI but also ongoing NVT FOI as well.*

Response: This has now been done. (Line: 504)

- *As the authors acknowledge, given the high NVT prevalence already in the community and the 2.5+ year surveillance occurred after vaccination it is very possible serotype replacement had already occurred (given the timeline is about 3 years). However, given the lack of pre-vaccination surveillance data this is not possible to validate.*

Response: we have addressed this point in the Limitations section. (Lines: 485-487)

- *Ultimately the biggest limitation to the paper and its findings is the lack of pre-vaccination data collected before Nov 2011 in this area. The authors wish to highlight that the program has not been as impactful in other settings at least in part to the 3+0 schedule used. However as we don't know what the original level of VT prevalence is it is somewhat difficult to confirm this using only their current data.*

Response: We are not disputing that PCV13 introduction has had an impact on VT carriage. However, we have shown that there is a high residual prevalence of VT carriage in vaccinated and unvaccinated populations which could lead to vaccine escape. Only when there is a formal evaluation of different schedules in this setting will we know whether this is due to the 3+0 schedule used. This has been clarified in the text. (Lines: 384-385; 390-394; 509-510)

References

- 1 Bar-Zeev N, Kapanda L, Tate JE, Jere KC, *et al.* Effectiveness of a monovalent rotavirus vaccine in infants in Malawi after programmatic roll-out: an observational and case-control study. *Lancet Infect Dis.* 2015;15:422-8.
- 2 Mvula H, Heinsbroek E, Chihana M, *et al.* Predictors of Uptake and Timeliness of Newly Introduced Pneumococcal and Rotavirus Vaccines, and of Measles Vaccine in Rural Malawi: A Population Cohort Study. *PLoS One* 2016;11: e0154997.
- 3 Sigaúque B, Moiane B, Massora S, *et al.* Early Declines in Vaccine Type Pneumococcal Carriage in Children Less Than 5 Years Old After Introduction of 10-valent Pneumococcal Conjugate Vaccine in Mozambique. *Pediatr Infect Dis J* 2018;37:1054–1060
- 4 Hammitt LL, Etyang AO, Morpeth SC, *et al.* Effect of ten-valent pneumococcal conjugate vaccine on invasive pneumococcal disease and nasopharyngeal carriage in Kenya: a longitudinal surveillance study. *Lancet* 2019; 393: 2146–54. [http://dx.doi.org/10.1016/S0140-6736\(18\)33005-8](http://dx.doi.org/10.1016/S0140-6736(18)33005-8)

Appendix – Abbreviated listing of changes in response to Reviewer#1 Manuscript NCOMMS-19-19575

Note: We have provided line numbers to show where text in the manuscript has been adjusted in response to reviewer’s comments. The line numbers refer to the Clean/Untracked versions of submitted Word files.

Abstract conclusions

- a. We have modified line 23 to minimise risk of misinterpretation (Lines: 23-24)
- b. We have modified line 24 minimise risk of misinterpretation (Lines: 24-27)

Background

- j. We updated the pneumococcal burden of disease figures, using a more recent publication by Wahl et al. (Lines: 35-37)
- k. We have provided further description of a previous Malawi carriage study (Lines: 97-100) as well as a table in the supplement (Supplementary Table 5). This shows the similarities in carriage dynamics between the Karonga and Blantyre study sites.
- l. We have provided additional references for HIC’s to support the statement that pre-PCV carriage prevalence in sSA differs “markedly from high-income settings”
- m. We have provided a range of VT carriage prevalence in HICs to compare to those estimates provide for LICs. (Lines: 64-66)
- n. We have aggregated ref 19 to ref list ‘15-18’ (Line: 70)
- o. We have added text to acknowledge the difficulty in interpreting data comparing countries and regions with a history of different PCV implementation strategies. We provide i) background on differences in PCV introduction (Lines: 47-56; 70-75) and ii) more detailed overview of the different vaccines and schedules that have been implemented (Supplementary Table 1).
- p. We have adjusted the text to clarify that NVT replacement is not unique to the Africa region. (Lines: 80-81)
- q. To ensure a more nuanced understanding that NVT carriage replacement does not necessarily mean disease replacement, we have adjusted the wording. (Lines: 81-84)
- r. To give more context to the discussion around a 3+0 schedule (Lines 60-62 in original submission), we have introduced text that compares PCV implementation in HICs (e.g. usually including booster doses) to the approach in LMICs in sSA (predominantly without booster doses). (Lines: 47-58)
- s. For external comparability we have provided further description of this study (Lines: 97-100) as well as a table in the supplement (Supplementary Table 5).

- t. We have further described HIV prevalence in this community (Lines: 90-95)
- u. We have adjusted text to ensure a consistent use of the term 'HIV-infected' when referring to the study's adult population.

Methods

- a. We have further described the questionnaire developed to assess PCV-vaccination status in the case a medical record was not available for confirmation. (Lines: 160-166)
- b. We have further addressed measurements of vaccine coverage, especially difficult to confirm in these field settings, in the Limitations section (Lines: 488-499)
- c. The prevalence ratio has been recalculated to show as an annual change. The description of these methods has been adjusted accordingly (Lines: 207-210)
- d. We have adjusted text in both the Methods (Lines: 223-230), the Results and the Supplementary material pages 10-12 (formerly Suppl 1) in order to help the readership better understand the modelling methods.
- e. We have adjusted text to ensure a consistent use of the term 'rolling cross-sectional surveys' when referring to the study design (Lines: Title, 9, 114, 284)
- f. We have provided additional context on activities that occurred before and during each survey that established these as individual rolling cross-sectional surveys. (Lines: 141-142, 148-149)

Results

- a. We report results of an analysis assessing whether individual covariates (as listed in Table 1) have changed over the study period. (Lines: 333-339)
- b. To assess impact of PCV's poor vaccine efficacy against ST3 carriage and disease, we have removed ST3 from the analysis and have reported the results in the supplementary material (Supplementary Table 9). We have included the results of this analysis, citing manuscript references reporting the poor efficacy of PCV13 against serotype 3. (Lines: 341-348)
- c. To address any bias due to age, we have restricted the ages of children included in the prevalence ratio (aPR) analyses, including 3-5-year-old PCV-vaccinated children and 6-8 year old PCV-unvaccinated children. We have described this in the Methods section (Lines: 211-216).
To provide the reader with the carriage data used in Figure 2, we have included the table submitted in our original submission (without the prevalence ratios) in Supplementary Tables 3 and 4.
- d. We clarified that the denominator for refusal rates were children screened (Lines: 258-263; footnote to the recruitment flow-chart [Figure-1]).

- e. We further described screening criteria, how eligible children were determined to be at home and who had authority to consent (Lines 145-153; footnote to recruitment flow-chart figure-1). We provided details on houses and individuals screened in the results section. (Lines: 258-264)
- f. While maintaining the use of survey number as primary unit of time sequence, we have introduced text to emphasise how this translates into 'time since Malawi's November 2011 introduction of PCV (3.6-7.1 years since PCV introduction).

Results, Table 1

- a. We added two additional variables: i) number of children <5yrs living in a household and ii) number of doses of PCV received.
- b. Further details on where children were recruited (household or school) was added as a footnote.
- c. We statistically compared variables in Table 1 (including household characteristics) between relevant groups. We report those that are statistically significant in the footnote of Table 1. We have described how the comparison was made in the Methods section (Lines: 203-205).

Discussion

- a. We have addressed vaccine coverage in the Limitations section (Lines: 488-499)
- b. To provide the context that the study occurred over a time period 3-7 years post PCV13, we have reiterated this point in the Abstract Results (Lines: 14-15), Results (Lines 241-242), Table-2 Footnotes, Discussion (Lines: 380-381).
- c. We have provided additional evidence from the literature to support the proposed mechanisms of vaccine-induced protection. (Lines; 405-410)

Supplement, Fig 2

- a. We have added a footnote to the figure clarifying a) the (0, 1] bar among unvaccinated children includes only children 4-8 weeks of age and b) the notation on x-axis '(' and ']'
- b. We have reported the denominators for each age band in Supplementary table 6

Supplement 2a, Table 1

- a. To report the Ns tested at each survey, we refer the reader to supplementary tables 3 and 4 which report 'Ns tested at each survey.' This is done in the supporting text for supplementary Tables 2a and 2b (now supplementary tables 7 and 8),

Figure 3

- a. *We have clarified that the y-axis is VT distribution, not prevalence.*
- b. We have clarified that the denominator of the percent is 'among VT isolates'

Figure 5

- a. We have adjusted the figure to show that PCV doses 1 & 2 do have some effect in controlling carriage.
- b. We have adjusted the figure to include Natural + Herd + Vaccine-induced immunities for the vaccinated arm. We have included Natural + Herd immunities for the unvaccinated arm.

Reviewers' Comments:

Reviewer #2:

None

Reviewer #3:

Remarks to the Author:

Results from this study demonstrate sustained circulation of vaccine-type pneumococci in a sub-Saharan African country several years after introduction of routine infant immunization with PCV13 using a 3 + 0 schedule, in a setting of high vaccine coverage. The manuscript is well-written and the data are presented clearly and will be of interest to the pneumococcal field. The revision addresses most of the previous reviewer comments and is strengthened by the inclusion of sensitivity analyses, assessment of the role of serotype 3 in the results, and more detailed description of the methods, particularly enrolment procedures and how vaccination status was determined. However, some of the conclusions regarding force of infection and waning immunity seem to be beyond the scope of the study objectives and therefore more appropriate as discussion points rather than conclusions. Specific comments, questions, and suggestions are provided below.

1. The relative reduction of VT carriage in unvaccinated 6-8 year olds (40.5%) and HIV+ adults (41.4%) was larger than the reduction observed in the vaccinated 3-5 year old children (16.1%). The authors note in lines 489-492 that readjustment of carriage dynamics may have already occurred between PCV13 introduction and the first carriage survey, which might explain the relatively minor changes seen in the vaccinated 3 – 5 year old age group. Could the authors discuss the results from 6-8 year olds and HIV+ adults in more detail, including why the decline in VT carriage might be more pronounced in these groups compared to vaccinated children? According to adjusted prevalence ratios, the largest change in VT carriage was seen in the HIV+ adult group, however lines 472 – 474 state “we now show that the HIV-infected adult population has not greatly benefitted from indirect protection against carriage...”

2. What exactly is meant by high force of infection (FOI) and how is this different than high carriage prevalence?

3. I feel that Figure 5 is too speculative; although it is intended to be a visualization of a hypothesis, it resembles a graph based upon a model, which might lead to misinterpretation. In particular, the 6 month point for an increase in VT carriage in vaccinated children, and the set-point of 12 months for naturally-acquired immunity are not sufficiently justified, and further exploration and modelling of this hypothesis (which would optimally utilize longitudinal data) seem to be outside the scope of this study. Similarly, although certainly appropriate for the discussion, I don't think conclusions should be made regarding waning immunity when this was not examined in the study.

4. Was enrolment evenly distributed across season? If not, adjusting for seasonality should be considered as seasonality was found to be associated with pneumococcal carriage in a previous study in Malawi (reference #16).

5. In addition to comparing carriage prevalence in survey 7 with survey 1, did the authors consider using a trend-over-time analysis that incorporated data from all years for examining changes in VT and NVT carriage over time?

Minor comments:

1. The correct name for GAVI is 'Gavi, the Vaccine Alliance'

2. In Table 2 and Supplementary Tables 3, 4, and 9, please report overall pneumococcal carriage rather than the proportion of participants with no carriage, as this will make it easier to compare

overall pneumococcal carriage rates with those reported in Supplementary Table 5 and the literature.
3. Supplementary tables 7 and 8 and Supplementary Figure 2 do not seem to be referred to in the manuscript text. In lines 380-381. It is not clear what is meant by Supplement 5, Figure 2.

Point-by-point response to the reviewer's comments manuscript NCOMMS-19-19575.

Note: We have provided line numbers to show where text in the manuscript has been adjusted in response to reviewer's comments. These lines refer to the 'Clean' manuscript.

Reviewer #3

Results from this study demonstrate sustained circulation of vaccine-type pneumococci in a sub-Saharan African country several years after introduction of routine infant immunization with PCV13 using a 3 + 0 schedule, in a setting of high vaccine coverage. The manuscript is well-written and the data are presented clearly and will be of interest to the pneumococcal field. The revision addresses most of the previous reviewer comments and is strengthened by the inclusion of sensitivity analyses, assessment of the role of serotype 3 in the results, and more detailed description of the methods, particularly enrolment procedures and how vaccination status was determined.

However, some of the conclusions regarding force of infection and waning immunity seem to be beyond the scope of the study objectives and therefore more appropriate as discussion points rather than conclusions. Specific comments, questions, and suggestions are provided below.

1. The relative reduction of VT carriage in unvaccinated 6-8 year olds (40.5%) and HIV+ adults (41.4%) was larger than the reduction observed in the vaccinated 3-5 year old children (16.1%). The authors note in lines 489-492 that readjustment of carriage dynamics may have already occurred between PCV13 introduction and the first carriage survey, which might explain the relatively minor changes seen in the vaccinated 3 – 5 year old age group.

Could the authors discuss the results from 6-8 year olds and HIV+ adults in more detail, including why the decline in VT carriage might be more pronounced in these groups compared to vaccinated children? According to adjusted prevalence ratios, the largest change in VT carriage was seen in the HIV+ adult group, however lines 472 – 474 state “we now show that the HIV-infected adult population has not greatly benefitted from indirect protection against carriage...”

Response: We have discussed this steeper decline among the older PCV-unvaccinated populations compared to 3-5 year olds. This includes a number of likely contributing factors, including i.) indirect benefits augmented by naturally acquired immunity and ii) evidence from a separate manuscript that the force of infection was found to be characterised by different transmission potentials within and between age groups. (Lines 279-289)

2. *What exactly is meant by high force of infection (FOI) and how is this different than high carriage prevalence?*

Response: The force of infection is the rate by which a certain age group of susceptible individuals is infected. We have included a reference and brief description of FOI, Lines 283-2784, in the context of addressing Point 1, above.

3. *I feel that Figure 5 is too speculative; although it is intended to be a visualization of a hypothesis, it resembles a graph based upon a model, which might lead to misinterpretation. In particular, the 6 month point for an increase in VT carriage in vaccinated children, and the set-point of 12 months for naturally-acquired immunity are not sufficiently justified, and further exploration and modelling of this hypothesis (which would optimally utilize longitudinal data) seem to be outside the scope of this study.*

Response: We have decided to remove Figure 5. If the Editor disagrees for any reason about its removal, we can replace the figure.

Similarly, although certainly appropriate for the discussion, I don't think conclusions should be made regarding waning immunity when this was not examined in the study.

Response: We have removed the mention of waning immunity other than in a sentence where we have postulated the role of waning immunity (Line 349-350)

4. *Was enrolment evenly distributed across season? If not, adjusting for seasonality should be considered as seasonality was found to be associated with pneumococcal carriage in a previous study in Malawi (reference #16).*

Response: Enrolment over the 7 surveys was evenly distributed across seasons. This has been emphasised in the manuscript (Lines 107-109).

5. *In addition to comparing carriage prevalence in survey 7 with survey 1, did the authors consider using a trend-over-time analysis that incorporated data from all years for examining changes in VT and NVT carriage over time?*

Response: As shown in table 2 and described in the Statistical Analysis (Lines 486-490), carriage prevalence ratios (PR) were calculated over the study duration by log-binomial regression using years (365·25 days) between date of Malawi's PCV introduction and participant recruitment, coded as a single time variable, allowing an estimate of (adjusted) prevalence ratio per annum.

Minor comments:

1. *The correct name for GAVI is 'Gavi, the Vaccine Alliance'*

Response: This has been adjusted

2. *In Table 2 and Supplementary Tables 3, 4, and 9, please report overall pneumococcal carriage rather than the proportion of participants with no carriage, as this will make it easier to compare overall pneumococcal carriage rates with those reported in Supplementary Table 5 and the literature.*

Response: We now report overall pneumococcal carriage rather than the proportion of participants with no carriage in Table 2 (now Table 3) and Supplementary Tables 3, 4, and 9

3. *Supplementary tables 7 and 8 and Supplementary Figure 2 do not seem to be referred to in the manuscript text. In lines 380-381. It is not clear what is meant by Supplement 5, Figure 2.*

Response: Reference to Supplementary tables 7 and 8 have been added to the manuscript (Lines 178-181). Reference to Supplementary Figure 2 has been added to the manuscript (Line 360). The reference to *Supplement 5, Figure 2* has been corrected to refer to *Supplementary figure 3 (Line 238)*